statistical physics/cognition/psychology

Zipf's Law, Herdan's Law, Brevity Law,
Menzerath–Altmann Law, lognormal distribution,
Buckeye corpus

**Authors for correspondence:**
Iván G. Torre
e-mail: ivan.gonzalez.torre@upm.es
Antoni Hernández-Fernández
e-mail: antonio.hernandez@upc.edu

# On the physical origin of linguistic laws and lognormality in speech

Iván G. Torre[1,2], Bartolo Luque[1], Lucas Lacasa[3], Christopher T. Kello[2] and Antoni Hernández-Fernández[4]

[1]Departamento de Matemática Aplicada, ETSIAE, Universidad Politécnica de Madrid, Plaza Cardenal Cisneros, 28040 Madrid, Spain
[2]Cognitive and Information Sciences, University of California Merced, 5200 North Lake Road Merced, 95343 CA, USA
[3]School of Mathematical Sciences, Queen Mary University of London, Mile End Road, E1 4NS London, UK
[4]Complexity and Quantitative Linguistics Lab, Laboratory for Relational Algorithmics, Complexity and Learning (LARCA), Institut de Ciències de l'Educació; Universitat Politècnica de Catalunya, Barcelona, Spain

IGT, 0000-0003-2380-010X; BL, 0000-0002-0396-4396;
LL, 0000-0003-3057-0357; CTK, 0000-0003-1588-9474;
AH-F, 0000-0002-9466-2704

Physical manifestations of linguistic units include sources of variability due to factors of speech production which are by definition excluded from counts of linguistic symbols. In this work, we examine whether linguistic laws hold with respect to the physical manifestations of linguistic units in spoken English. The data we analyse come from a phonetically transcribed database of acoustic recordings of spontaneous speech known as the Buckeye Speech corpus. First, we verify with unprecedented accuracy that acoustically transcribed durations of linguistic units at several scales comply with a lognormal distribution, and we quantitatively justify this 'lognormality law' using a stochastic generative model. Second, we explore the four classical linguistic laws (Zipf's Law, Herdan's Law, Brevity Law and Menzerath–Altmann's Law (MAL)) in oral communication, both in physical units and in symbolic units measured in the speech transcriptions, and find that the validity of these laws is typically stronger when using physical units than in their symbolic counterpart. Additional results include (i) coining a Herdan's Law in physical units, (ii) a precise mathematical formulation of Brevity Law, which we show to be connected to optimal compression principles in information theory and allows to formulate and validate yet another law which we call the size-rank law or (iii) a mathematical derivation of MAL which also highlights an additional regime where the law is inverted. Altogether, these results support the hypothesis that statistical laws in language have a physical origin.

# 1. Introduction

The so-called linguistic laws—statistical regularities emerging across different linguistic scales (i.e. phonemes, syllables, words or sentences) that can be formulated mathematically [1]—have been postulated and studied quantitatively over the last century [1–5]. Notable patterns which are nowadays widely recognized include Zipf's Law which addresses the rank-frequency plot of linguistic units, Herdan's Law (also called Heaps' Law) on the sublinear vocabulary growth in a text, the Brevity Law which highlights the tendency of more abundant linguistic units to be shorter, or the so-called Menzerath–Altmann Law (MAL) which points to a negative correlation between the size of a construct and the size of its constituents.

Despite the fact that spoken communication pre-dates written communication, the vast majority of studies on linguistic laws have been conducted using written corpora or transcripts [6,7]—to the neglect of oral communication—with some notable exceptions [8–11]. As a matter of fact, linguistics and cognitive science are traditionally based on a foundation of symbolic representation. For instance, Harley states that language itself is 'a system of symbols and rules that enable us to communicate' [12], and Chomsky assumes that the symbolic nature is presupposed to construct linguistic models [13]. Chomsky goes even further, adding that 'it is tacitly assumed that the physical signal is determined, by language-independent principles, from its representation in terms of phonetic symbols' [13, p. 107]. In some sense, this perspective intends to construct their linguistic models focusing on symbols, giving more credit to the visual communication underlying writing than the orality and the acoustic origin of language—as if symbolism preceded acoustics. Under such a paradigm [14] that we could term as the *symbolic hypothesis*, the above-mentioned statistical laws would emerge in language use as a consequence of its symbolic representation.

However, language use also has important non-symbolic aspects like variations in acoustic duration, prosody and speech intensity, which carry non-verbal information complementing the (purely symbolic) transcribed text [15] with well-known acoustic implications in e.g. clinical linguistics [16]. For instance, a given word or sentence can be spoken in different ways, with different intonations, and therefore its duration admits a certain variability [8] that could have semantic consequences [17]. These variations cannot be explained—by construction—using symbolic language representations, and therefore one would not expect physical measures to follow the linguistic laws without an additional explanation.

To address this important issue, here we have conducted a systematic exploration of linguistic laws in a large corpus of spoken English (Buckeye corpus) [18,19] which has been previously manually segmented, hence having access at the same time to both (i) symbolic linguistic units (the transcription of phonemes, words and breath-groups (BG), defined by pauses in the speech for breathing or longer and (ii) the physical quantities attached to each of these units, which altogether allow a parallel exploration of statistical patterns of oral communication in both the actual physical signal and its text transcription.

We first explore the time duration of linguistic units at several scales and are able to verify with unprecedented accuracy that these systematically comply with a lognormal distribution (LND). This apparently universal regularity—which we might even call a *lognormality law*—is then justified in the light of a simple stochastic model that is able to explain quantitatively the onset of LNDs at word and BG linguistic scales just assuming lognormality at the phoneme scale.

In a second step, we address the parallel investigation of classical linguistic laws in oral communication in both the actual acoustic signal and its text transcription. We certify that the classical Zipf's Law emerges in oral transcribed communication at word and phoneme level, establishing that we are facing a 'standard' corpus. We then find that Herdan's Law holds in physical magnitudes of time duration and we are able to analytically link the exponent of this law with the one found for the case of symbolic units. We subsequently show that Zipf's Law of abbreviation also holds in spoken language, and to the best of our knowledge we obtain for the first time experimental evidence of an exponential law dependency between the frequency of a linguistic element and its size, a relation which we mathematically explain invoking information-theoretic arguments [20]. This new mathematical formulation of Zipf's Law of abbreviation in turn enables the mathematical formulation of yet another law relating the size and the rank of words.

Notably, such patterns are boosted when measuring size using physical magnitudes (time duration) rather than written magnitudes (number of phonemes or characters). This emphasis is even stronger for the MAL, which we show to hold better only if size of linguistic units is measured in physical terms (time duration) rather than in symbolic units. We also include a model that explains the origin of this fourth law.

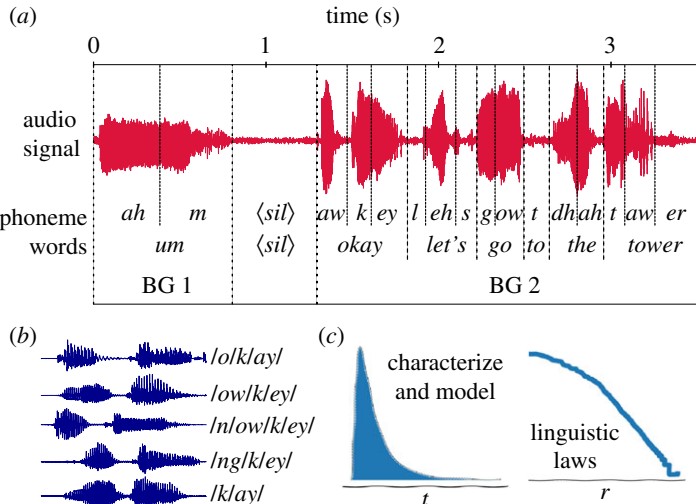

**Figure 1.** Sketch of the database and analysis. (*a*) We show the waveform of a speech sample and the alignment for three linguistic levels of symbolic transcription: phonemes, words and BG. (*b*) We showcase how the same symbolic unit (the word *okay*) may show a wide diversity within speech communication (number of phonemes, phoneme type, duration, etc.). We combine all this information in order to (*c*) characterize statistical patterns and linguistic laws in both symbolic and physical magnitudes at three different levels, and discuss the relationship between them.

We end up discussing the relevance of each of our results and finally briefly analyse the implication of these on the validity of the symbolic hypothesis versus a novel 'physical hypothesis', and the consequences of this potential paradigm shift within theoretical linguistics.

## 2. Material and methods

The so-called Buckeye corpus database contains conversational speech by native English speakers gathering approximately $8 \times 10^5$ phonemes, $3 \times 10^5$ words and $5 \times 10^4$ BGs with time-aligned phonetic labels [18,19,21]. Recordings are based on interviews with 40 native central Ohio speakers and balanced for age, gender and gender of interviewer (interviews are essentially monologues of each interviewee), and technical details on the phonetic segmentation are reported in the electronic supplementary material.

Accordingly, we had access to speech recordings segmented with their symbolic transcriptions at the phoneme and word levels. The corpus also included transcriptions of pauses that we used to define a third, larger unit of analysis, roughly corresponding to the so-called BG. BGs are typically defined by pauses in the speech for breathing or longer [22], a fundamental quantity, for example, in the study of verbal fluency [23]. While one can *a priori* assume that punctuation in written texts could drive pauses and therefore help to define BGs directly from written texts, such an issue is not so clear in spontaneous communication, and in general BGs cannot be directly inferred from transcribed speech. Transcribed breaks in the Buckeye corpus included pauses, silences, abrupt changes in pitch or loudness, or interruptions [18,19]. Each segmented unit included physical magnitudes such as time onset and time duration.

We then use this manual segmentation to make a parallel analysis of linguistic patterns based on (classical) symbolic units and—when possible—complementing those with analysis of the respective patterns based on physical (time duration) magnitudes. In figure 1, we depict an example for illustration purposes, where we show a manually segmented word (okay) which is described by standard linguistic measures such as the precise list of phonemes composing it. The particular nature of oral communication sometimes allows a given word to be composed by different sets of phonemes (figure 1 shows several *different* phonetic transcriptions of the word okay found in the corpus). Note that this source of variability is by construction absent in written texts and clearly enriches oral communication. On top of this, note that the same word can be spoken with different time duration along speech, due to different factors including prosody, conversational context, etc. [8]. For instance, the word okay is found a number of times over the corpus, and for each event we annotate its time duration. We can therefore estimate a time distribution for this word, from which a mean or median value can be extracted. In general, every phoneme, word and BG which is manually segmented has an associated time duration, hence empirical time duration distributions can be estimated for these

**Table 1.** Parameters across linguistic levels. Number of elements considered (N), mean, standard deviation (s.d.), mode, median and percentiles 10 (p10) and 90 (p90) of a physical magnitude (time duration distribution) versus symbolic ones (number of characters, number of phonemes and number of words) for the three linguistic levels (phoneme, words and BGs). Since speakers sometimes omit or add phonemes to the same word, the number of characters per phoneme is obtained indirectly averaging number of phonemes and number of characters in the word. The p10 and p90 percentiles give us an account of the range of durations, without considering outliers.

| | | time duration $t$ (seconds) | | | | | |
|---|---|---|---|---|---|---|---|
| | N | mean $\langle t \rangle$ | s.d. | mode | median | p10 | p90 |
| phoneme | $8 \times 10^5$ | 0.08 | 0.06 | 0.05 | 0.07 | 0.03 | 0.14 |
| words | $3 \times 10^5$ | 0.24 | 0.17 | 0.12 | 0.2 | 0.08 | 0.45 |
| BG | $5 \times 10^4$ | 1.4 | 1.2 | 0.4 | 1.1 | 0.3 | 3.1 |
| | | mean | s.d. | mode | median | p10 | p90 |
| number of characters | | | | | | | |
| phoneme | | 1.4 | 0.5 | 1 | 1.3 | 2 | 2 |
| words | | 4 | 2 | 4 | 4 | 2 | 7 |
| BG | | 24 | 23 | 2 | 17 | 3 | 54 |
| number of phonemes | | | | | | | |
| words | | 3 | 1.6 | 2 | 3 | 1 | 5 |
| BG | | 18 | 17 | 2 | 13 | 2 | 40 |
| number of words | | | | | | | |
| BG | | 6 | 6 | 1 | 4 | 1 | 13 |

different linguistic scales. The mean, mode and median of the time duration distribution of phonemes, words and BGs are reported in table 1. Taking advantage of the segmentation and alignment of the oral corpus with the corresponding text [18,19], we have also computed the statistics referring to the number of characters of each linguistic level, of phonemes per word and BG, and of words per BG.

As we will show in the next section, the probability distributions of phoneme, word and BG size (in time duration or other magnitudes) are heavy-tailed (more concretely subexponential in the classification of Voitalov *et al.* [24]) so the mean or the standard deviation are not necessarily informative enough, this is why we also report the most frequent value (mode) and the median. Due to the inherent uncertainties in the segmentation and the known existence of outliers, extreme cases are better represented by percentiles 10 and 90 than by the minimum and maximum values. See the electronic supplementary material for a thorough discussion on how basic speech metrics collected in this corpus compare with the ones found in other works.

The number of words per BG, of phonemes per word and BG, and of characters per phoneme, word and BG are also depicted in table 1. The fact that the most common BG is formed by a single word—with the dubious element `um` as the most frequent—influences our results (i.e. the mode of the number of phonemes per word and per BG is 2), reflecting nevertheless the characteristics of spontaneous speech where discursive markers abound: they are key elements in verbal fluency, many of which are brief linguistic elements (`so`, `okay`, `well`, etc.) [25]. Furthermore, the conditions of the Buckeye corpus are interview-like conversational speech (where interviewer makes questions and the analysis is then performed on the interviewee): this significant trait probably makes the abundance of dubious elements [25] (e.g. `um`) large.

# 3. Results

## 3.1. Lognormality law

Here we analyse the marginal distribution of the physical magnitude under study: the time duration each segmented linguistic unit, at all scales (phonemes, words and BGs). For a given linguistic level—e.g. words—we measure the time duration of all events (different words and repetitions) found in the

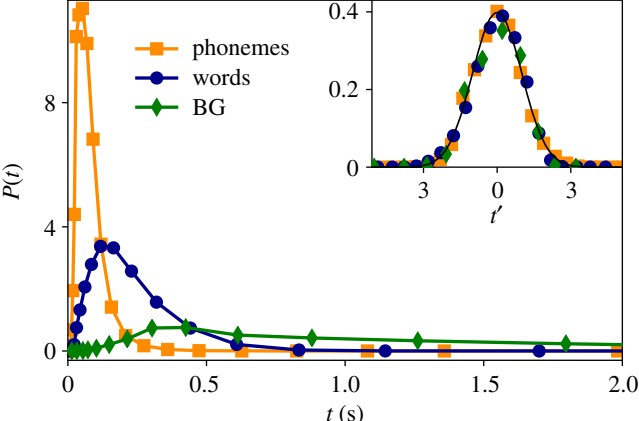

**Figure 2.** Universal lognormal time duration distribution. (Outer panel) Estimated time duration distribution of all BGs (green diamonds), words (blue circles) and phonemes (orange squares) in the English Buckeye database (a logarithmic binning has been applied to the histograms, and solid lines are guides for the eye). In each case, the curves fit well to a LND (see the text and table 2 for a model selection). (Inner panel) We check the validity of the lognormal hypothesis by observing that, when rescaling the values of each distribution $t' = (\log(t) - \langle\log(t)\rangle/\sigma(\log(t)))$, all data collapse into a universal standard Gaussian (solid black line is $\mathcal{N}(0, 1)$).

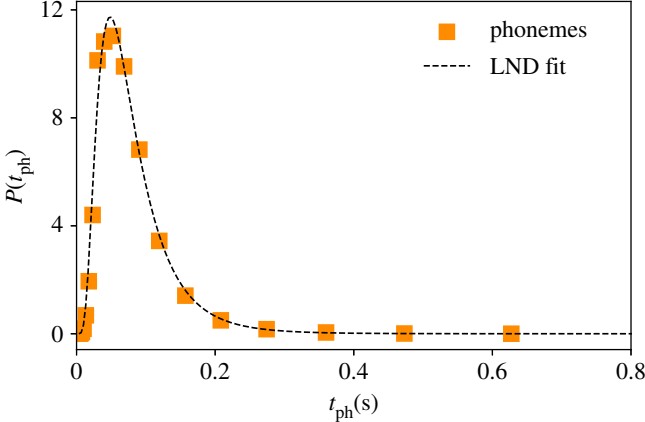

**Figure 3.** Phoneme duration is lognormally distributed. Empirical time duration distribution of phonemes (orange squares) in the English Buckeye database. Black dotted line is a maximum likelihood estimation (MLE) fit to a LND, with MLE parameters (−2.68, 0.59) (see table 2 for goodness of fit and alternative fits).

corpus. That is to say, we do not use the time average of each word, but consider each event as a different sample. In the main panel of figure 2, we then show the time duration distributions for phonemes (orange squares), words (blue circles) and BG (green diamonds) in the Buckeye corpus (see also figures 3–5). Using the method of maximum-likelihood estimation (MLE) [26], we have fitted the data to five possible theoretical distributions: lognormal (LND), beta, gamma, Weibull and normal (we use Kolmogorov–Smirnov distance $D_{ks}$ for goodness of fits, and mean loglikelihood for model selection, see table 2). We have confirmed that both phonemes and BG are best explained by LNDs

$$\text{lognormal}(x; \mu, \sigma) = \frac{1}{x\sigma\sqrt{2\pi}} \, e^{-((\ln(x)-\mu)^2/2\sigma^2)},$$

whereas for the case of words, LND, beta and gamma are similarly plausible statistical fits. In the inset panel of figure 2, we re-scale all the time duration variables $t' = (\log(t) - \langle\log(t)\rangle/\sigma(\log(t)))$. If all distributions are well described by LNDs, the resulting data should collapse to a standard Gaussian $\mathcal{N}(0, 1)$, in good agreement with our results. Note at this point that the Buckeye corpus is multi-speaker, hence data come from a variety of speakers. Nevertheless, the lognormality law still holds for individual speakers (see the electronic supplementary material).

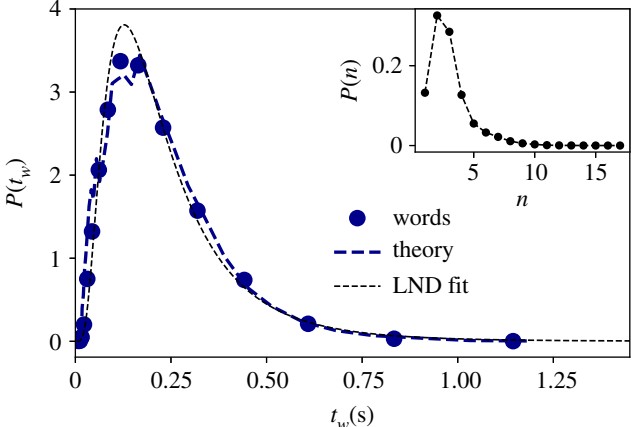

**Figure 4.** Word duration is lognormally distributed. Empirical time duration distribution of words (blue circles) in the English Buckeye database. Black dotted line is the MLE fit to a LND distribution (see table 2 for goodness of fit). Blue dashed line is the theoretical prediction of equation (3.1) (see the text for details). (Inset panel) Estimated distribution of number of phonemes per word $P(n)$ from which $n$ is randomly sampled.

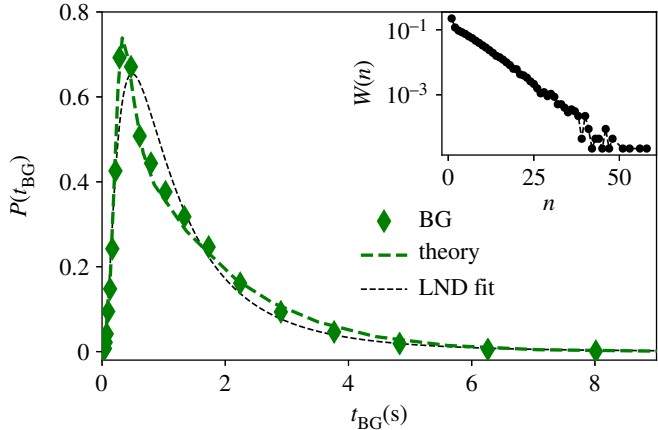

**Figure 5.** BGs are lognormally distributed time duration distribution of all BGs (green diamonds) in the English Buckeye database. Black dotted line is the MLE fit to a LND distribution (see table 2 for goodness of fit). The green dashed line is the theoretical prediction of equation (3.1) where a Gaussian error term with positive mean is included to model for voice onset time (VOT; see the text for details). (Inset panel) Semi-log plot of the estimated distribution of number of words per BG $W(n)$ from which $n$ is randomly sampled in the theoretical model. Note that this distribution is exponential, suggesting that BG segmentation of the corpus is statistically compliant with a Poisson process.

LNDs are indeed very commonly found across natural and behavioural sciences [27–29], and it is well known [30,31] that the frequency distribution of string length in texts (in both letters and phonemes) is well approximated at every level by the LND (see [28] and references therein). Previous studies have proposed that LND is indeed consistent for spoken phonemes in several languages [30,32–35], and this distribution has also been found, although overlooked, in the distribution of word durations for English [8]. However, to the best of our knowledge this is the first study in which LNDs have been reported at various linguistic levels at the same time.

Can we justify the onset of clear LNDs for the time duration of phonemes, words and BGs? To date, most of the theoretical work connecting the presence of LND for the time duration of linguistic units reduce to an extremely vague analogy and reminiscence of stochastic multiplicative processes and the central limit theorem (CLT) in logarithmic space [27,32]. The mechanistic origin for the robust duration distribution of phonemes is therefore an open problem which we would not address here, although it could be speculated that this is a consequence of some underlying physiological or cognitive process [36]. We now assume LND for the time duration of phonemes as a working hypothesis (nonetheless validated by the experimental evidence reported in figure 2), and we provide a (mechanistically justified) mathematical model that explains why, in that case, both words and BG should have a

**Table 2.** Estimated LND parameters ($\mu$, $\sigma$) for time duration distributions of phonemes, words and BG. Distribution candidates are fitted using MLE and model selection is based on maximizing the mean log-likelihood $\langle \mathcal{L}(\widehat{\theta}) \rangle$. On the right side of the table, we depict Kolmogorov–Smirnov goodness-of-fit distance $D_{ks}$ for different alternative distributions (data are more plausible to follow the distribution with lower values of $D_{ks}$). All tested distributions are defined with two parameters and inside each category level are evaluated in same conditions.

| | LND | | model selection $\langle \mathcal{L}(\widehat{\theta}) \rangle$ | | | | | goodness of fit $D_{ks}$ | | | | |
| --- | --- | --- | --- | --- | --- | --- | --- | --- | --- | --- | --- | --- |
| | $\mu$ | $\sigma$ | LND | beta | gamma | Weibull | normal | LND | beta | gamma | Weibull | normal |
| phoneme | −2.68 | 0.59 | 1.8 | 1.76 | 1.76 | 1.69 | 1.48 | 0.014 | 0.052 | 0.05 | 0.081 | 0.128 |
| word | −1.62 | 0.66 | 0.63 | 0.63 | 0.63 | 0.61 | 0.45 | 0.015 | 0.014 | 0.018 | 0.035 | 0.099 |
| BG | 0.025 | 0.86 | −1.29 | −1.31 | −1.31 | −1.32 | −1.64 | 0.036 | 0.047 | 0.045 | 0.047 | 0.13 |

duration which themselves approximates a LND, which we show to be not just qualitatively correct but also offers an excellent quantitative agreement with the empirical results.

### 3.1.1. A simple stochastic model

Consider a random variable (RV) $Y \sim \text{lognormal}(\mu, \sigma)$ that models the time duration $t_{\text{ph}}$ of a given phoneme. Since words are constructed by concatenating a certain number of phonemes $n$, the duration of a given word $t_w$ can then be modelled as another RV $Z$ such that

$$Z = \sum_{i=1}^{n} Y_i, \tag{3.1}$$

where we assume $Y_i \sim \text{lognormal}(\mu, \sigma)$ and $n \sim P(n)$ is in general yet another RV. For the sake of parsimony, we initially consider the case of independent RVs: how is $Z$ distributed when the RVs $Y_i$ and $n$ are sampled independently? Since the lognormal distribution has finite mean and variance, the CLT should hold and $Z$ should be Gaussian as $n \to \infty$. Interestingly, this is a limit theorem and thus the Gaussian behaviour is only deemed to be recovered in the limit of large $n$. However, this is quite not the case in our context: not is only $n$ a finite yet fluctuating RV, furthermore, according to table 1, the average number of phonemes per word is just $\langle n \rangle_{\text{phon}} = \langle t_w \rangle / \langle t_{\text{ph}} \rangle = 0.24/0.08 \approx 3$, whereas in the case of the average number of words per (oral) BG, we find $\langle n \rangle_{\text{words}} \approx 6$, both in principle sufficiently far from the large $n$ limit where CLT holds in the lognormal case (see the electronic supplementary material for an exploration). While for small $n$, there is no closed form for $P(Z)$ in the general case, it is agreed that the CLT does not kick in [37] and actually the LND is often a good approximation for $P(Z)$ (see [38] and references therein), and one can approximate the first two moments of $Z$ using e.g. the so-called Fenton–Wilkinson approximation [39]. We have numerically checked that this is indeed the case provided that $Y_i$ are sampled from reasonably similar LNDs (see the electronic supplementary material for details). In other words, this simple stochastic model can already explain the emergence of LND for the duration of words solely based on the assumption that phoneme durations also follow a LND. Subsequently, one can redefine $Y_i = t_w$ with the time duration of a word—which now is justified to follow a LND—and $Z = t_{\text{BG}}$ with the time duration of a BG, hence this very same model also explains the emergence of LNDs of BG durations.

Moreover, in order to be *quantitatively* accurate, instead of sampling $n$ from a synthetic probability distribution we can sample it from the actual distribution of phonemes per word $P(n)$ (reported in the inset panel of figure 4). In other words, in order to construct words according to equation (3.1), each $Y_i$ is sampled from the phoneme time distribution $P(t_{\text{ph}})$, whereas the number of phonemes per word $n$ is sampled from the real distribution $P(n)$ instead of a synthetic one. The results of this version of the model is plotted, for the case of words, as a dashed blue curve in figure 4, finding an excellent quantitative agreement with the empirical distribution.

One could proceed to do a similar exercise for the case of BGs, where the number of words per BG $n$ is an RV which is sampled from the actual distribution $W(n)$, as reported in semi-log scales in the inset of figure 5. Note, incidentally, that $W(n)$ is exponentially decaying, suggesting that the segmentation of BGs is statistically analogous to a (memoryless) Poisson process. In any case, such procedure is, in this case, problematic: observe that manual segmentation tends to have a systematic error which is more prominent in the case of BGs due to the fact that one needs to determine the transition points between speech and silence (i.e. errors do not cancel out in this case.[1]) This is indeed known to be a non-trivial problem due to the so-called VOT effect at the beginning of some BGs and other phonetic phenomena, possibly amplified by the fact that manual segmentation tends to be conservative (see the electronic supplementary material for details). These sources of error will thus systematically add a small positive bias to the true time duration of each BG. Thus, we decide to model this bias by a Gaussian error term with (small) positive mean, which is systematically added to the time duration RV $Z$, so that $Z \to Z + \xi$, where $\xi \sim \mathcal{N}(\mu_\xi, \sigma_\xi)$. In the main panel of figure 5, we report the prediction of this model when $\xi \sim \mathcal{N}(0.14, 0.07)$ (green dashed curve), showing excellent agreement with the empirical distribution (note that $\mu_\xi$ and $\sigma_\xi$ can safely vary within a 20% range and the agreement would still be remarkable).

### 3.1.2. Tackling non-independence

The stochastic model discussed above is already able to quantitatively reproduce the time duration distributions of words and BGs, even if we assumed that the RVs $Y_i$ and $n$ were independent. This is,

---

[1]Note that segmentation errors might take place when segmenting words as well; however, in this case they tend to cancel out—what is erroneously added to one word is removed from the subsequent segmented word—and thus these errors have zero mean.

however, a rather strong assumption which is not true in general: it is easy to see that if these were independent, then e.g. MAL should not hold. Possible sources of interaction between RVs include dependence between $n$ and $Y_i$ and serial correlations between $Y_i$ and $Y_{i+1}$. To assess the case of serial correlations, we have estimated the mutual information $I(t, t+1)$ between duration of subsequent linguistic units for both phonemes inside a word and words inside a BG. The mutual information $I(X_1, X_2)$ is an information-theoretic measure that evaluates lack of independence by quantifying how much information is shared by two RVs $X_1$ and $X_2$:

$$I(X_1, X_2) = \sum_{x_i \in X_1} \sum_{x_j \in X_2} p(x_i, x_j) \log \left( \frac{p(x_i, x_j)}{p(x_i)p(x_j)} \right), \tag{3.2}$$

such that $I(t_1, t_2) \to 0$ if $t_1$ and $t_2$ are independent. In practice, finite-size effects prevent this quantity from vanishing exactly, so a robust analysis requires comparing the numerical estimate with respect to a proper null model. Note that we use here $I$ instead of other methods such as Pearson or Spearman correlation coefficients because we cannot assume *a priori* any particular dependence structure (such as linear correlation or monotonic dependency).

We found $I(t_1, t_2)_{\mathrm{phon}} = 3 \times 10^{-2}$ and $I(t_1, t_2)_{\mathrm{words}} = 2 \times 10^{-2}$ for phonemes and words, respectively, to be compared with the results for a null model where we keep fixed the number of words and phonemes per word but we shuffle the phoneme allocation, hence breaking possible correlations. We find $I^{\mathrm{rand}}(t_1, t_2)_{\mathrm{phon}} = 3 \times 10^{-4} \pm 2 \times 10^{-5}$ and $I^{\mathrm{rand}}(t_1, t_2)_{\mathrm{word}} = 2 \times 10^{-4} \pm 3 \times 10^{-5}$ for phonemes and words, respectively. In both cases, mutual information is two orders of magnitude stronger than what is expected due to chance, safely concluding that the RVs $Y_i$ in equation (3.1) are indeed *not* independent. In the section devoted to Menzerath–Altmann, we further examine the properties of these correlations for the case of words, and in the electronic supplementary material, we exploit these to build an independent model that also accounts for experimental time duration distribution of BGs once we add such dependence structure.

Importantly, as opposed to the previous case, there do exist limit theorems for $Z$ and small $n$ when $\{Y_i\}$ are *not* independent. A theorem of Beaulieu [40] states that the probability distribution of a sum of positively correlated lognormal RVs having a specified correlation structure approaches a LND with probability one, so we safely conclude that equation (3.1) provides a sound justification for the emergence of LND regardless of the underlying correlation structure. Incidentally, this limit theorem is also valid for some joint lognormal RVs having dissimilar marginal distributions, as well as identically distributed RVs, hence we do not require that the intraphonemic time duration of all phonemes be identical for this theorem to hold (we acknowledge that intraphonemic variability exists but leave this fine-grained structure for a future work).

## 3.2. Zipf's Law

We now turn to explore the emergence of linguistic laws in oral communication, and start our analysis with Zipf's Law. After some notable precursors [41–43], George Kingsley Zipf formulated and explained in [44,45] one of the most popular quantitative linguistic observations known in his honour as Zipf's Law. He observed that the number of occurrences (frequency $f$) of words with a given rank $r$ is well approximated by a power-law dependence

$$f(r) \sim r^{-\alpha}. \tag{3.3}$$

This is a solid linguistic law proven in many written corpora [5] and in spoken language [6], even though its variations have been discussed [46], as is the case of the evolution of the exponent in the ontogeny of language [7] or even in aphasia [47].

Zipf originally proposed a theoretical exponent $\alpha \sim 1$ [45], but other authors have shown that $\alpha$ may vary for oral English typically between 0.6 and 1.5 [6,7] (note that the actual fitting of a power law is not trivial, and different proposals coexist [48,49], here we use MLE). Other authors have further justified the existence of two different scaling regimes: one kernel of very versatile communication elements and one set of almost unlimited lexicon [50,51], whereas other authors support that this is an artefact of mixing texts [52].

Here, we analyse the written transcriptions of the Buckeye corpus, and summarize our results for the frequency-rank word plot in figure 6, finding that our results in spontaneous conversation agree with previous studies. We indeed find that a double power law scaling should be preferred from a model selection point of view according to a Bayesian information criterion (BIC, see the electronic supplementary material for details), with exponents $\alpha_1 \sim 0.63$ and $\alpha_2 \sim 1.41$ with breaking point in

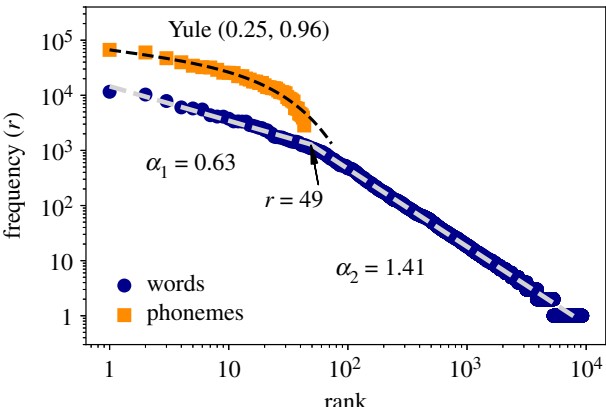

**Figure 6.** Zipf's Law. Log–log frequency-rank plots of phonemes (orange squares) and words (blue circles). We have found two regimes of power law for the case of words with exponents ($\alpha_1 = 0.63$, $\alpha_2 = 1.41$) and breaking point at rank $r = 49$. We fitted phonemes rank distribution to a Yule distribution $f(r) \sim r^{-b} c^r$ with parameters $b = 0.25$, $c = 0.96$. All fittings (including estimation of the breaking point) were performed using MLE.

rank $r = 49$ (the precise breaking point is found using MLE). We also find, in compliance with Williams *et al.* [52], that when we disentangle contributions from different speakers, the double power law seems to smear out (see the electronic supplementary material).

The relationship between the exponent of Zipf's Law and syntax has been discussed previously [7,53]. Accordingly, the exponents found in the Buckeye corpus would be in the range expected for a low syntactic complexity, typical of spontaneous speech, with a predominance of discursive markers [25], although more research is needed in this regard.

In figure 6, we have also analysed Zipf's Law at phoneme level. While the limited number of phonemes precludes the onset of distributions ranging over one decade—and therefore limits the interpretability of these results—some previous studies [54] have stretched the analysis and proposed the onset of Zipf's Law in phoneme distributions, proposing a fit of this frequency-rank plot in terms of a Yule distribution $f(r) \sim r^{-b} c^r$ (note that a power-law distribution is a particular case of Yule distribution which can also be explained as a power law with exponential cut-off). Accordingly, we have fitted the transcribed phonemes to a Yule distribution using MLE, finding $b = 0.25$, and $c = 0.96$.

## 3.3. Herdan's Law

We now move to the second linguistic law under study. Although with little-known precedents [55], Herdan's Law [56] (also known as Heaps' Law [57]) states that the mean number of new different words $V$ grows sublinearly with the size of the text $L : V \sim L^{\beta}$, $\beta < 1$. Interestingly, some scholars have derived an inverse relationship between Zipf's and Herdan's exponents $\beta = 1/\alpha_2$ using different assumptions (see [58] or [59] for a review). As Zipf's Law, Herdan's Law is robust although there are slight deviations that have been well explained by new formulations [6,58–60].

Here we explore the emergence of Herdan's Law by measuring the appearance of new words as conversations draw on using either total number of words $L$ (classical approach) and elapsed time $T$, and we report our results in figure 7. Since the corpus is multi-author, we have performed several permutations of the order in which the corpus concatenates each of the individual speakers, and plot each of these permutations as a different line (10 permutations). Results hold independently of the permutation so we can rule out that such arbitrary ordering is playing any role in the specific value of the exponents.

Green diamonds depict the increase of vocabulary as a function of the total number of words appearing in the conversation as it draws on. For words, we find a first linear regime where each word is new, followed up by a transition into a stable sublinear regime that holds for about three decades with exponent $\beta \approx 0.63$. This evidence is in agreement with previous results [6]. Note that the exponent $\beta$ is approximately consistent with the one found for the second regime of Zipf's Law ($1/1.41 \approx 0.7$) and others reported for different corpora [6,58,61].

In the same figure, we also depict (blue circles) the vocabulary growth as a function of the *time elapsed* $T$, i.e. we count how many new words appear in a given conversation as the conversation draws on, and

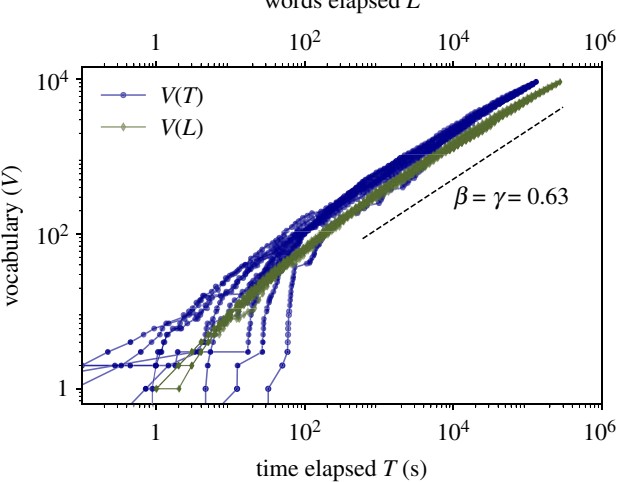

**Figure 7.** Herdan's Law. Sublinear increase of number of different words ($V$) used during spontaneous speech versus time elapsed $T$ (blue circles) and total number of words spoken $L$ (green diamonds), where each line is the result of a different permutation in the way of concatenating the different speakers of the multi-author corpus (10 permutations). After a transient range, we find a robust scaling regime $V \sim L^{\beta}$ and $V \sim T^{\gamma}$ for about three decades. The exponents coincide $\beta = \gamma \approx 0.63$ (see the text for a justification).

we measure the size of the vocabulary $V$ as a function of the elapsed time $T$. Note that this formulation is strongly related with the speech rate of conversation which might vary greatly between speakers, context or intentionally of the speaker. Whereas here we find that the transient is dependent on the specific permutation of speakers, all curves then transition into a permutation-independent, stable and robust sublinear regime $V \sim T^{\gamma}$ with approximately the same exponent $\gamma \approx \beta \approx 0.63$.

This later result can be explained analytically in the following terms. Consider equation (3.1) and concatenate a total of $L$ words, each having a duration modelled by a RV $Y$ (which we know is lognormal according to previous sections). Assuming there are no silences between words, the concatenation variable $\tau = \sum_{i=1}^{L} Y$ is a RV that can be identified with the elapsed time of a conversation after $L$ words. The average time $T = \mathbb{E}(\tau)$ and, since the expected value is a linear operator, it follows that $T = \sum_{i=1}^{L} \mathbb{E}(Y) = \mathbb{E}(Y) \cdot L$ (note that taking expected values is justified when $L$ is large by virtue of the law of large numbers, see electronic supplementary material, figure S7 for an empirical validation). Since we find $T \propto L$, this implies that if Herdan's Law holds for $L$, a similar law with the *same* exponent should hold for $T$, i.e. $\beta = \gamma$.

## 3.4. Brevity Law

The third linguistic law under analysis is Brevity Law, also known as Zipf's Law of abbreviation. It *qualitatively* states that the more frequently a word is used, the 'shorter' that word tends to be [44,45,62]. Word size has been measured in terms of number of characters, according to which the law has been verified empirically in written corpora from almost a thousand languages of 80 different linguistic families [63], and similarly logograms tend to be made of fewer strokes in both Japanese and Chinese [64,65]. The law has also been observed acoustically when word size is measured in terms of word time duration [8,66,67], and recent evidence even suggests that this law also holds in the acoustic communication of other primates [68]. Despite all this empirical evidence, and while the origin of Zipf's Law of abbreviation has been suggested to be related to optimization principles [45,69–71], to the best of our knowledge the law remains qualitative and a precise mathematical formulation is lacking.

Here, we start by studying Brevity Law qualitatively in oral speech at the level of phonemes and words (it is not possible to check at BG level due to lack of statistics).

At word level, we consider three cases: (i) the tendency of more frequent words to be composed of less characters, (ii) the tendency of more frequent words to be composed by a smaller number of phonemes and finally, (iii) the tendency of speakers to articulate more frequent words in less time (i.e. the more frequent a word, the shorter its duration). Results are summarized in figure 8, showing that Brevity Law indeed holds in all three cases. In these figures, we scatter-plot the frequency of each word versus the three different definitions of word size (median time duration, number of phonemes, number of characters). Blue dots are the result of applying a logarithmic binning over the frequencies

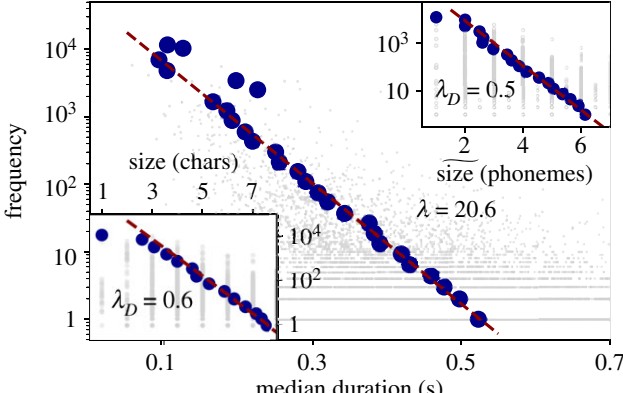

**Figure 8.** Brevity Law: words. On the main panel, we display for each word (grey dots) the scatter plot between its median time duration (in seconds) and its frequency of occurrence. Blue circles are the result of applying logarithmic binning to frequencies (see the text and electronic supplementary material for details). Upper right panel shows the same relationship but considering the median number of phonemes per word, while bottom left panel represents the number of characters per word. All panels are semi-log plots. Spearman test $S$ systematically shows strong negative correlation slightly higher for time duration ($S \sim -0.35$) than for symbolic representation of phonemes and characters ($S \sim -0.26$) (a two-sided $p$-value for a hypothesis test whose null hypothesis is that two sets of data are uncorrelated provides $p$-value $\ll 10^{-3}$ in all the cases, i.e. safely rejecting the null hypothesis of uncorrelation), supporting the Zipf's Law of abbreviation in every case. Data in the inset panels are then fitted to theoretical equation (3.4), whereas data in the main panel is fitted using theoretical equation (3.5). In every case, there is a remarkable agreement between the theoretical equations (which fit the grey dot data) and the binned data.

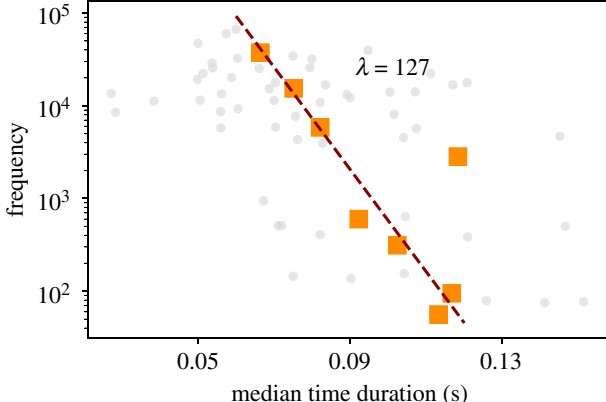

**Figure 9.** Brevity Law: phonemes. Scatter plot between the frequency of each phoneme and its median time duration (grey dots). Orange squares are the result of applying a logarithmic binning over frequencies (see the electronic supplementary material for details). Spearman correlation test shows a weak negative correlation ($S \sim -0.2$) with $p$-value $p < 0.05$, but strong negative correlation ($S \sim -0.5$) and $p$-value $< 10^{-3}$ when we only consider the subset phonemes with frequencies $f > 50$.

axis in order to counterbalance low sampling effects (see the electronic supplementary material for additional details on this procedure).

At the phoneme level, we compare the frequency of phonemes with their median time duration in figure 9. As suggested by Spearman correlation test, we find that Zipf's Law of abbreviation holds even at such a low linguistic level. In this way, the more frequent is a phoneme, the shorter it will be in terms of duration. We have not addressed the law at the phoneme scale with respect to the number of characters as this assignation is ambiguous due to the fact that a given word can have different phonetic transcriptions (figure 1a), and it is not obvious how to assign the number of characters to the phoneme composition of each of these.

### 3.4.1. A mathematical formulation of Brevity Law

An information-theoretic principle of compression [20] has been recently invoked to elucidate the origin of Zipf's Law for the frequency of words [72]. A similar approach can be undertaken here. In order to

compress information, a classical rule of thumb is to codify information by assigning shorter labels to more frequent symbols (e.g. the most frequent symbol shall be assigned the shorter label, the second most frequent symbol shall be assigned the second shorter label, and so on). The size of this label is called the *description length*, which here we associate with either of the three possible definitions of size we have considered (median time duration, number of phonemes and number of characters). In information-theoretic terms, if a certain symbol $i$ has a probability $p_i$ of appearing in a given symbolic code with a $\mathcal{D}$-ary alphabet, then its minimum (optimal) expected description length $\ell_i^* = -\log_{\mathcal{D}}(p_i)$ [20]. Deviating from optimality can be effectively modelled by adding a pre-factor, such that the description length of symbol $i$ is $\ell_i \sim -(1/\lambda_{\mathcal{D}})\log_{\mathcal{D}}(p_i)$, where $0 < \lambda_{\mathcal{D}} \leq 1$. Identifying $p_i$ with the frequency of a given word and $\ell$ with its 'size', the derivation above directly yields a mathematical formulation of Zipf's Law of abbreviation as

$$f \sim \mathcal{D}^{-\lambda_{\mathcal{D}}\ell}, \quad 0 < \lambda_{\mathcal{D}} \leq 1, \tag{3.4}$$

where $f$ is the frequency of a linguistic element, $\ell$ is its size in whichever units we measure it (some property of the time duration distribution, number of phonemes and number of characters), $\mathcal{D}$ is the size of the alphabet (the number of different linguistic elements at the particular linguistic level under study), and $\lambda_{\mathcal{D}}$ an exponent which quantifies deviation from compression optimality (the closer this exponent is to one, the closer to optimal compression).

A fit to equation (3.4) is shown (red dashed lines) in the upper right and lower left inset panel of figure 8. When word size is measured in number of characters (i.e. the alphabet consists of letters and thus $\mathcal{D} = 26$), we find $\lambda_{\mathcal{D}} \approx 0.6$, whereas for word size measured in terms of number of phonemes (i.e. for an alphabet with $\mathcal{D} = 64$ phonemes consisting in 41 phonemes plus 23 phonetic variations including flaps, stops and nasals, see the electronic supplementary material), we find $\lambda_{\mathcal{D}} \approx 0.5$. Note that both fits are performed to the data (not to the binned data), but these are in turn in excellent agreement to the binned data (blue circles).

On the other hand, when word size is measured in terms of time duration, there is no natural alphabet, so $\mathcal{D}$ is *a priori* not well defined (time is a continuous variable). We can nonetheless express equation (3.4) as

$$f \sim \exp(-\lambda\ell), \quad \lambda > 0 \tag{3.5}$$

where $\lambda$ is now just a fitting parameter not directly quantifying the distance to optimal compression, and $\ell$ is some measure of 'centrality' of the time duration distribution. In the main panel of figure 8, we plot (red dashed line) a fit of equation (3.5) to the data when $\ell$ is measured in terms of the median time duration, finding $\lambda \approx 20.6$ (again, the fit is performed to the noisy data cloud, but the result is in excellent quantitative agreement to the binned data). A similar fit to the case of phonemes is presented in figure 9.

### 3.4.2. Connecting Brevity Law and Zipf's Law: the size-rank law

Zipf and Herdan Laws are known to be connected and under certain conditions their exponents are related via $\alpha = 1/\beta$ [59,60]. Now since Zipf's Law and the newly formulated Brevity Law involve word frequencies, we can now connect these to propose an additional law. Putting together equations (3.3) and (3.5), our theory predicts that the 'size' $\ell_i$ of word $i$ is related with its rank $r_i$

$$\ell_i = \frac{\alpha}{\lambda}\log(r_i) + K = \theta\log(r_i) + K, \tag{3.6}$$

where $\alpha$ and $\lambda$ are Zipf and Brevity Laws exponents, respectively, and $K$ a normalization constant. $\theta$ is therefore a parameter combining Zipf and Brevity exponents in a size-rank plot, and equation (3.6) can indeed be understood as a new linguistic law by which the larger linguistic units tend to have a higher rank following a logarithmic relation.

In the case of double power-law Zipf Laws (figure 6), we would have different exponents for $r \geq 50$ or $r < 50$ so equation (3.6) would reduce to

$$\left.\begin{array}{l} \ell_i = \theta_1\log(r_i) + K_1, \quad \text{if} \quad r_i \leq r^*, \\ \ell_i = \theta_2\log(r_i) + K_2, \quad \text{if} \quad r_i > r^*, \end{array}\right\} \tag{3.7}$$

where $\theta_1 = (\alpha_1/\lambda)$, $\theta_2 = (\alpha_2/\lambda)$, and $\alpha_1$ and $\alpha_2$ are the exponents before and after the breaking point $r^*$, respectively. We illustrate the validity of equation (3.7) by considering time duration as $\ell$. In this scenario, $\lambda \approx 20$, $\alpha_1 \approx 0.63$ and $\alpha_2 \approx 1.41$, so $\theta_1 \approx 0.03$ and $\theta_2 \approx 0.07$. In figure 10, we depict a $r$ versus $\ell$

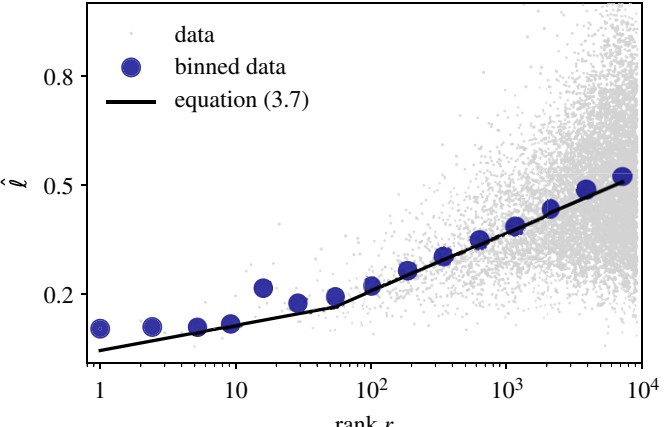

**Figure 10.** Size-rank law for words. Linear-log scatter plot of the median time duration $\ell$ of each word as a function of its rank $r$ (grey dots). Blue circles are the same data after a logarithmic binning. Black line is equation (3.7) with $\theta_1 = (\alpha_1/\lambda) = 0.03$ and $\theta_2 = (\alpha_2/\lambda) = 0.07$, where $\alpha_1$, $\alpha_2$ and $\lambda$ are the two Zipf's exponents and Brevity exponent, respectively.

scatter plot of all the words, where blue dots are the result of applying a logarithmic binning (again, to counterbalance low sampling effects). The black line is equation (3.7) with $\theta_1 = 0.03$ and $\theta_2 = 0.07$, showing an excellent agreement with the binned data.

## 3.5. Menzerath–Altmann Law

To round off, we finally consider the fourth classical linguistic law. After some precedents in experimental phonetics [73], Paul Menzerath experimentally observed a negative correlation between the 'length' of a phonetic constructs and the length of its constituents [10,11]. Later Gabriel Altmann formalized this observation for various linguistic levels [74,75], proposing a mathematical formulation called MAL, which in its most popular form relates the size $n$ of a language construct (the whole) and the size $y$ of its constituents $y$ (the parts) via

$$y(n) = an^b \exp(-cn), \tag{3.8}$$

where $a$, $b$, $c$ are free parameters that depend on language [5,76] (see also [77,78] for subsequent attempts of reformulation).

The interpretation and justification of this formulation remains unclear [76], and while this law was originally explored phonetically [10], most of the works address written texts [1,68,76,79–82].

### 3.5.1. Two different regimes

As an initial comment, note that when both exponents $b$, $c < 0$, equation (3.8) has always a finite minimum at $n^* = b/c$ above which the tendency inverts, i.e. the law would be a decreasing function for $n < n^*$ and an increasing function for $n > n^*$, leading in this latter case to a 'the larger the whole, the *larger* the size of its constituents' interpretation. This rather trivial observation seems to have been unnoticed, and the right end of MAL's standard formulation has been systematically overlooked in the literature—perhaps due to the fact that this regime is hard to find experimentally—even if Menzerath himself already observed this tendency in his seminal works [10,11].

### 3.5.2. A mechanistic model for Menzerath–Altmann's Law

Second, and before addressing to which extent MAL holds in oral communication, we now advance a model that provides a *mechanistic* origin for its precise mathematical formulation. Let $t(n)$ be the average time duration of a construct (BG) formed by $n$ constituents (words). Then the mean duration of a word inside that construct is $y(n) = t(n)/n$. Let us assume that a BG formed by $n$ words can be generatively explained by adding a new word to a BG formed by $n-1$ words. Under no additional information, one can trivially express that the new word has a duration equivalent to the average

duration of words in the BG, i.e.

$$t(n) = t(n-1) + \frac{t(n-1)}{n-1} = (1 + \frac{1}{n-1})t(n-1). \tag{3.9}$$

This defines a simple multiplicative process which can be solved as

$$t(n) = t(1) \prod_{j=2}^{n} (1 + \frac{1}{j-1}) = nt(1),$$

yielding $y(n) = t(1)$, i.e. a constant quantity. We can call this the 'order-0 approximation' to MAL. Now, in §3.1 we found that word time duration is indeed correlated within a BG, as the mutual information between the duration of a given word and the subsequent one is much larger than expected by chance. One can take into account this correlation in several ways. The simplest way is to assume a linear relation by which the size of the $n$th constituent of a construct is just a constant fraction $0 < \kappa_2 < 1$ of the average of the previous $n-1$ constituents (i.e. the size of the construct grows slower than linearly with the number of constructs due to linear correlations). Then equation (3.9) is slightly modified into

$$t(n) = t(n-1) + \kappa_2 \frac{t(n-1)}{n-1} = (1 + \frac{\kappa_2}{n-1})t(n-1), \tag{3.10}$$

such that

$$t(n) = t(1) \prod_{j=2}^{n} (1 + \frac{\kappa_2}{j-1}).$$

The expression above is in total agreement with eqn 12 in [78], although the authors in [78] do not solve this equation and simply propose it as a 'formula'. Now it is easy to see using gamma functions that

$$\prod_{j=2}^{n} (1 + \frac{\kappa_2}{j-1}) = \frac{\Gamma(n+\kappa_2)}{\Gamma(1+\kappa_2)\Gamma(n)},$$

where $\Gamma(z)$ is the gamma function. Invoking the fact that $\forall \alpha \in \mathbb{C}$, $\lim_{n \to \infty} \Gamma(n+\alpha)/[\Gamma(n)n^\alpha] = 1$, we can approximate in this case

$$y(n) = \frac{t(n)}{n} \sim \frac{t(1)}{\Gamma(1+\kappa_2)} n^{\kappa_2-1}, \tag{3.11}$$

which for $\kappa_2 < 1$ is a decaying power-law relation, sometimes called the restricted MAL [83]. This would be the 'order-1 approximation' to MAL.

We can continue the procedure and in the next level of simplicity ('order-2'), we can add another pre-factor $\kappa_1$, such that the generative model reads then

$$t(n) = \kappa_1 (1 + \frac{\kappa_2}{n-1})t(n-1),$$

such that

$$t(n) = t(1) \prod_{j=2}^{n} \kappa_1 (1 + \frac{\kappa_2}{j-1}). \tag{3.12}$$

Note that if $\kappa_1 < 1$ we can risk eventually finding an average time duration $t(n)$ smaller than $t(n-1)$, which is unphysical, so a safe assumption is setting $\kappa_1 \geq 1$. While equation (3.12) does not have an easy closed-form solution, we can analytically approximate it. By taking logarithms and Taylor-expanding $\log(1 + \kappa_2/(j-1)) \approx \kappa_2/(j-1)$, equation (3.12) reads

$$\log t(n) \approx \log(t(1)\kappa_1^{n-1}) + \sum_{j=2}^{n} \frac{\kappa_2}{j-1}.$$

Using harmonic numbers $H_n$ [84], we have

$$\sum_{j=2}^{n} \frac{\kappa_2}{j-1} = \kappa_2 \left( H_n \frac{-1}{n} \right) \sim \log n^{\kappa_2} + \kappa_2 \gamma + O\left( \frac{1}{n} \right),$$

where $\gamma = 0.5772\ldots$ is the Euler–Mascheroni constant. Putting these results altogether, taking

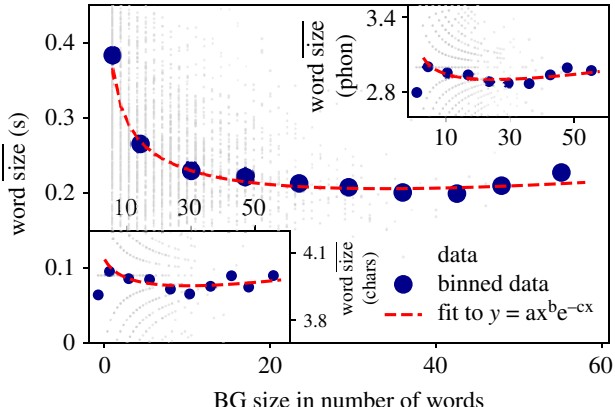

**Figure 11.** Menzerath–Altmann Law: BG versus words. Relation between BG size (measured in number of words) versus the mean size of those words. The mean size of words can be expressed in terms of mean time duration (main panel), mean number of phonemes (upper right panel) or mean number of characters (lower left panel). Each grey point represents one BG, whereas blue circles are the mean duration of BGs (for exposition purposes, a linear binning has further been applied to smooth out the data and reveal the emergence of two different MAL regimes; equivalent plots without this additional binning are reported in the electronic supplementary material). Red dotted line is a fit to MAL (equation (3.8)). The law clearly holds only when constituent size is measured in physical units (table 3).

exponentials and using $y(n) = t(n)/n$, we end up with

$$y(n) \approx \frac{t(1)\exp{(\kappa_2 \gamma)}}{\kappa_1} n^{\kappa_2 - 1} \kappa_1^n,$$

which is indeed MAL (equation (3.8)) with $a = t(1)\exp(\kappa_2\gamma)/\kappa_1$, $b = \kappa_2 - 1$ and $c = -\log\kappa_1$. Note that since $0 < \kappa_2 < 1$, then necessarily $b < 0$, and since we had set $\kappa_1 \geq 1$, that also means that $c < 0$, and therefore we expect in full generality that MAL displays its two regimes.

Furthermore, the model we provide not only gives a mechanistic interpretation for the origin of equation (3.8), but also shows that actually two parameters ($\kappa_1$, $\kappa_2$) are enough to fit the law instead of three, and these two parameters quantify the way correlations between the duration of words take place.

As an additional comment, note that if instead of equation (3.12) we decide to model correlations by exponentiating by a factor $\kappa_2$, i.e.

$$t(n) = t(1)\prod_{j=2}^{n}\kappa_1(1 + \frac{1}{j-1})^{\kappa_2}, \tag{3.13}$$

then this equation is exactly solvable as $\prod_{j=2}^{n}(1 + 1/[j-1])^{\kappa_2} = [\Gamma(n+1)/\Gamma(n)]^{\kappa_2} = n^{\kappa_2}$, hence in this latter case there is no approximation and we find

$$y(n) = \frac{t(1)}{\kappa_1} n^{\kappa_2 - 1} \kappa_1^n,$$

i.e. again equation (3.8) with $a = t(1)/\kappa_1$, $b = \kappa_2 - 1$, $c = -\log(\kappa_1)$. Notice, however, that equation (3.13) is probably harder to interpret than equation (3.12) (see the electronic supplementary material, figure S7 for a successful prediction of BGs time duration distribution solely based on the models above).

### 3.5.3. Menzerath–Altmann's Law is fulfilled better in physical units

Once the origin of equation (3.8) has been clarified, we now explore to what extent MAL holds in oral communication at two linguistic levels: (i) BG versus word and (ii) word versus phoneme. For case (i), we measure the size of each BG in terms of number of words and then compare this quantity against the size of the constituents (words) using three different measures: (a) mean number of characters in the constituent words, (b) mean number of phonemes in the constituent words and (c) mean time duration of the constituent words. Accordingly, cases (i.a) and (i.b) relate different linguistic levels, whereas (i.c) provides a link with quantities which are inherently 'oral'.

Results for the BG versus word scale are shown in figure 11. In all cases, we have plotted each individual instance of a BG as grey dots, and blue circles correspond to linear binned data. For the

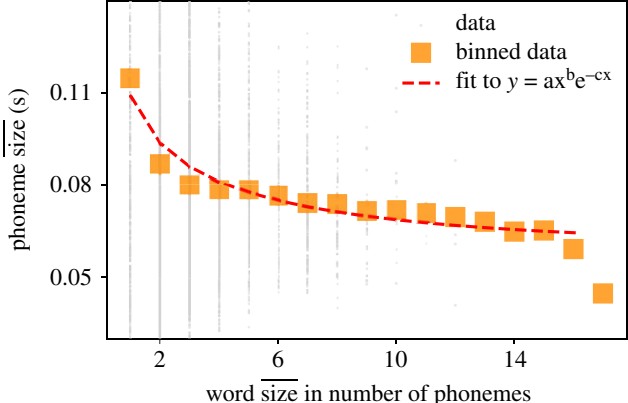

**Figure 12.** MAL: words–phonemes. We show the relation between word size (measured in number of phonemes) and the mean time duration of those phonemes. Orange squares are mean duration of words with that size. Each grey point represents one word and dotted line is a fit to MAL (equation (3.8)).

**Table 3.** Parameter fits of the Buckeye corpus to MAL (equation (3.8)) for different linguistic levels (BG, words and phonemes). Fitting of MAL to the mean values (mean size of constituent versus mean size of linguistic construct) has been done using Levenberg–Marquardt algorithm (note that blue circles in figure 11 are the result of a linear binning). $R^2$ (coefficient of determination) is used to determine the goodness of the fit. Accordingly, MAL only holds most significantly when measuring constituent size in time units.

|  | $a$ | $b$ | $c$ | $R^2$ |
|---|---|---|---|---|
| BG versus word size (in time units) | 0.364 | $-0.227$ | $-6.7 \times 10^{-3}$ | 0.7 |
| BG versus words (in number of phonemes) | 3.22 | $-4.7 \times 10^{-2}$ | $-1.85 \times 10^{-4}$ | 0.05 |
| BG versus word size (in number of chars) | 4.16 | $-2.14 \times 10^{-2}$ | $-7.4 \times 10^{-4}$ | 0.05 |
| words versus phoneme size (in time units) | 0.18 | $-0.23$ | $-7 \times 10^{-3}$ | 0.9 |

sake of exposition, we have further applied a linear binning of the data with 10 bins (see the electronic supplementary material for the more standard plots using non-binned data and additional details). This additional binning helps to visually reveal the emergence of the second MAL regime. The red dashed line is a fit of the data to equation (3.8) using a Levenberg–Marquardt algorithm (see table 3 for fitting parameters). We find that MAL between a construct and its constituents is only shown to hold significantly when the constituent size is measured in physical (time) units according to $R^2$ (when the law is measured in symbolic units, one could even say that the order-0 approximation provided by equation (3.9) is the adequate model). We find $b, c < 0$ so equation (3.8) indeed is non-monotonic in this case and, interestingly, the law fits indeed the whole range including the regime where the interpretation of MAL inverts, with a transition located at a BG size $b/c \approx 34$ words.

For case (ii), we ensemble words with the same number of phonemes and then compute mean time duration of those phonemes; see figure 12 for results. Again in this case MAL is found to hold.

### 3.5.4. Average speech velocity

Finally, observe that when word size is measured in time duration, $y(n)$ in equation (3.8) is indeed the average time duration per word in a BG with $n$ words, hence we can define an *average speech velocity* $v(n) = n/[ny(n)] = 1/y(n)$. Different speakers will therefore have different speech velocity, and this can also vary along their speech. Now, for the range of parameters where $y(n)$ is non-monotonic, $v(n)$ will be non-monotonic as well, and the critical point $n^* = b/c$ which fulfils $v'(n^*) = 0$ defines, assuming MAL, a maximal limit (optimal efficiency limit) for the number of words per second. For the fitted parameters found in the corpus, we plot $v(n)$ in figure 13, finding an optimal efficiency limit at around 4.8 words per second, achieved when BGs last about $n^* \approx 34$ words. The *average* number of words in a BG in the Buckeye corpus is, however, around 6 (for which the speech velocity is only about 4 words per second), meaning that on average speakers chat more slowly. We can imagine a number of reasons why speakers

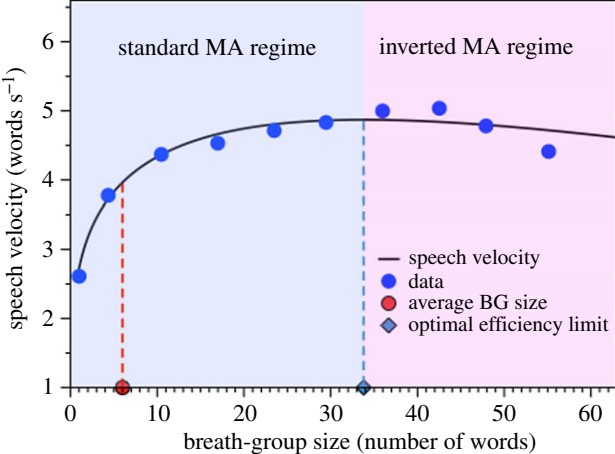

**Figure 13.** Speech velocity function. When word size is measured in time units, $v(n) = 1/y(n)$, where $y(n)$ is MAL (equation (3.8)) defines the speaker's speech velocity. This function is in good agreement with empirical data (blue circles) and is maximal for BGs with $n^* \approx 34$ words. Experimentally, however, we observe BGs with an average size of about 6 words, hence suggesting a suboptimal speech velocity. The critical point $n^*$ indeed separates a classical regime $n < n^*$ where the standard interpretation of MAL holds, and an inverted regime $n > n^*$ where the interpretation of the law switches.

in the Buckeye corpus have on average a slightly suboptimal efficiency, from obvious ones where conversational speech is performed in a relaxed environment where the speakers do not need to optimize information transmission per unit time, to more speculative ones where lung capacity (which imposes a physiological limit to BG size) plays a limiting role.

# 4. Discussion

Linguistic laws—traditionally explored in the context of written communication—are validated here with unprecedented accuracy within oral communication, both in the acoustic space spanned by time duration and in symbolically transcribed speech. Since oral communication pre-dates written communication, it is sensible to wonder whether the emergence of linguistic laws in the latter context are indeed just a consequence of their emergence in the former. In that case, the question of why and how these complex patterns first emerged in orality would directly point towards investigating how the cognition and physiology of human beings evolved according to, among other gradients, an evolutive pressure driven by human interaction and their need to communicate. These questions also suggest the need to perform comparative studies [68,85] that explore the emergence of similar complex structures in the oral communication of other species, a task which is theoretically possible even if the underlying language code is unknown [86].

It is now worth discussing the breadth and depth of our specific results. The first one deals with the time duration distribution of linguistic units in oral communication. We have certified that the time duration of phonemes, words and BGs in our database are lognormally distributed, and coined this universal character as the *lognormality law*. Furthermore, we were able to mechanistically explain these traits at the word and BG level through the introduction of a stochastic generative model, which is able to hierarchically explain the time duration distribution at a certain linguistic level from the one emerging at the linguistic level immediately below. Remarkably, this model also predicts the correct quantitative shape of the word and BG time duration distributions, and the sole assumption we make is that phonemes themselves are also lognormal, a hypothesis that we empirically validate and is supported by previous analysis [8]. Note that lognormality of phonemes has been discussed previously in the context of multiplicative processes [87] and Markov chains [32]; however, we consider that a sound explanation for its origin is still lacking, an issue—which we speculate is a by-product of underlying physiological processes [36]—that is left as an open problem for future work. Finally, note that our models do not require a multi-speaker context: while the Buckeye corpus is multi-speaker, individual analyses are also in agreement with the model (see the electronic supplementary material).

On a second step, we turned to investigate more traditional linguistic laws in the context of oral communication, and started with Zipf's Law. Our results for this case, where we find two scaling

regimes (figure 6), in agreement with [6,50] and in line with [52], which claims that double power-law scalings are expected in multi-author corpus (see the electronic supplementary material for a clarification of this aspect). Since each word can be spoken in different ways (e.g. with different time durations and energy release), it is not obvious how to explore Zipf's Law in physical units; however, see [86].

Then we turned to Herdan's Law, where we found that the standard law $V \sim L^{\beta}$ holds also in oral communication, and that a *newly* defined one—where we use accumulated time elapsed $T$ (in seconds) instead of total number of words $L$—holds as well and with the same exponent, an observation that we were able to analytically justify. These findings reinforce the idea that statistical patterns observed in written language naturally follow from analogous ones emerging in oral speech. As a detail, note that the transition towards the stable regime relates to the fact that the Buckeye corpus consists of concatenating multi-author corpus and therefore requires a speaker-dependent transient until the system reaches a stable state, as previously found in some other studies [6].

Subsequently, we considered the third classical linguistic law: the Brevity Law or Zipf's Law of abbreviation, where shorter words tend to be used more frequently. To the best of our knowledge, here we introduce for the first time an explicit mathematical formulation of this law (equations (3.4) and (3.5)) which we justify based on optimal compression principles [20,72]. This information-theoretic formulation predicts that the law should emerge in both symbolic and physical magnitudes. We were able to show that this law (and its novel mathematical formulation) indeed holds in oral communication when measured both in the traditional setting (using number phonemes or characters as a proxy for word size) and when using physical magnitudes (using time duration to measure word size). Since both Brevity and Zipf's Law address the frequency of linguistic units, we have also been able to mathematically connect them to propose a new size-rank law which we have also shown to hold (figure 10).

The principle of optimal compression provides yet another reason why patterns in the acoustic setting underpin the more commonly observed ones in written texts: on average a word is composed by a small number of phonemes or characters because this word—which is a symbolic transcription of a spoken word—was originally spoken fast (short time duration), not the other way around. All the above notwithstanding, and although the tendency to Brevity is probably a universal principle [69], we should also acknowledge that in certain communicative contexts it may not be fulfilled if there are other conflicting pressures such as sexual selection [88], noise [89], long-distance calls [9] or other energetic constraints [90].

We finally addressed MAL in oral communication at different scales (BG versus words, and words versus phonemes). We first were able to derive a mechanistic model based on the concatenation of words that mathematically explains the onset of MAL as proposed by Altmann. The law itself mathematically predicts a second regime where the popular MAL interpretation is inverted, and empirical results support the presence of this second regime here (additional analysis in other datasets should confirm whether the onset of the second regime is generally appearing, or whether this is just a mathematical artefact).

Interestingly, we find that MAL is fulfilled when the constituent size is measured in physical units (time duration, $R^2 = 0.7$ and 0.9) but notably less so when we use the more classic written units ($R^2 = 0.05$ and 0.05, table 3).

This is yet another indirect indication supporting that written corpus is a discrete symbolization that only captures the shadow of a richer structure present in oral communication [8] in which linguistic laws truly emerge, with a physical origin that written texts would only capture partially. As a matter of fact, working in time units enabled us to also define a speech velocity function, which we have found to be slightly below the optimal efficiency limit imposed by the actual MAL, a deviation which was indeed expected considering the fact that conversational speech is not under stress to maximize informational content per unit time. Note at this point that MAL has been traditionally argued to emerge only in linguistic units lying on adjacent levels [91], and under the symbolic perspective words would not be considered the element immediately below the BG, and similarly the phoneme is not the element immediately below the word (i.e. the seminal work of Menzerath and De Oleza related words with syllables [10]). Nonetheless, working with physical units (time durations) MAL is fulfilled both between the BG and word levels, and between the word and phoneme levels.

It should also be noted that, to the best of our knowledge, this is the first study relating acoustic levels such as BGs and words. As a matter of fact, assuming that MAL has a purely physiological origin [10,11] that eventually percolated into a written corpus, then BGs should indeed be more adequate units to explore this law than sentences or clauses. BGs are free from some of the problems emerging for sentences and clauses [92], and actually are so universal that they can also be measured in animal communication [68].

To conclude, we have thoroughly explored and explained a number of statistical patterns emerging in oral communication, which altogether strongly suggest that complexity in written texts—quantitatively summarized in the so-called linguistic laws—is not necessarily an inherent property of written language (the symbolic hypothesis) but is in turn a by-product of similar structures already present in oral communication, thereby pointing to a deeper, perhaps physiological origin [36,93]. As a contrast to the symbolic hypothesis, we coin this new perspective as the *physical hypothesis*. The extent by which the physical hypothesis holds has been previously certified in prelinguistic levels [86] and to some extent in phonology [94,95] and ecological psychology [96]. In the framework of linguistic laws, we argue that these must be studied primarily by analysing the acoustic magnitudes of speech, since their recovery in written texts is due to the extent to which they collapse and are a reflection of orality. In Chomskyan terms, our results suggest that linguistic laws come from non-symbolic principles of language performance, rather than symbolic principles of language representation. Also, we believe the physical hypothesis allows a more objective analysis within theoretical linguistics—including the classical debate on linguistic universals—and avoids many espistemological problems [97]. For instance, this paradigm does not negate the symbolism of language, much on the contrary it ultimately aims at explaining its origin without any needs to postulate it *a priori*.

Further work should be carried out to confirm our findings, including similar analysis for additional physical variables such as energy dissipation, and comparative studies (i) in other languages and (ii) in oral communication for other species [86].

Data accessibility. Buckeye corpus is a freely accessible corpus for non-commercial uses. Post-processed data from Buckeye corpus is now available in Dryad Digital Repository: https://doi.org/10.5061/dryad.4ss043q [98], while scripts are now available in https://github.com/ivangtorre/physical-origin-of-lw. We have used Python 3.7 for the analysis. Levenberg–Marquardt algorithm, Kolmogorov–Smirnov distance, Spearman test and most of MLE fits use Scipy 1.3.0. MLE fit for power law is self-coded. Other libraries such as Numpy 1.16.2, Pandas 0.24.2 or Matplotlib 3.1.0 are also used.

Authors' contributions. All authors designed the study. I.G.T. performed the data analysis. B.L. and L.L. contributed analytical developments. All authors wrote the paper.

Competing interests. We declare we have no competing interests.

Funding. I.G.T. acknowledges support from Programa Propio (Universidad Politécnica de Madrid), Fulbright-Schuman program and the hospitality of CogSci Lab, UC Merced. I.G.T. and B.L. were supported by the grant FIS2017-84151-P (Ministerio de Economia, Industria y Competitividad, Gobierno de España). L.L. acknowledges support from EPSRC Early Career Fellowship EP/P01660X/1. A.H-F. was supported by the grant no. TIN2017-89244-R (MACDA) (Ministerio de Economia, Industria y Competitividad, Gobierno de España) and the project PRO2019-S03 (RCO03080 Lingüística Quantitativa) de l'Institut d'Estudis Catalans.

Acknowledgements. The authors thank extensive comments from anonymous referees.

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
