## [Reviewer comments · Royal Society Open Science]

Review History

RSOS-191023.R0 (Original submission)

Review form: Reviewer 1

Is the manuscript scientifically sound in its present form?

Yes

Are the interpretations and conclusions justified by the results?

Yes

Is the language acceptable?

Yes

Do you have any ethical concerns with this paper?

No

Have you any concerns about statistical analyses in this paper?

Yes

Recommendation?

Accept with minor revision (please list in comments)

Comments to the Author(s)

I am very much in favor of the publication of this paper. It is not only a great but an ambitious paper. The authors study the fulfilment in recorded speech of the 4 main laws of quantitative linguistics. In fact, they include an additional law, which describes the probability distribution of durations of words, phonemes, and breath groups (I do not know why they only mention “four laws” then). So, the scope of the paper is very broad, which is welcome. However, this broadness can be also a weak point, as the reader may find that some additional explanations and/or investigations can be necessary, to reach an in-depth treatment of each law. I am not in favor of “salami publication”, but the point is that, the more things in the paper, the more likely the referee finds something he/she doesn’t like.

My point is that this paper has to be considered a sort of “overview” and that in future research the authors can extend their results. Nevertheless, it would be necessary that the authors mention in the manuscript that extensions and more in-depth analysis can be done, leaving the issue as an open one. So, it can be good for the readers to know that the research can be extended, and that the authors show the way to do that. It is within this philosophy that some of my comments below should be understood.

Most relevant points:

- 1. IIIA: Time durations: It is not totally clear what the authors are measuring. Please clarify in manuscript. Are they providing the distribution of all the individual events in the corpus (e.g., 3e5 words)? Or the distribution of the mean value of each unique event (e.g., dictionary words)? Or is the median value instead of the mean? In any case, the former has of course greater variability. In other words, how are the events weighted? Any of these distributions is of interest, and they are related.
- 2. Related to this, my understanding is that time durations mix different speakers. Which is the effect of the velocity of every speaker? In other words, does the skewness come from the fact that they look at a diverse population of speakers? A single speaker also leads to a lognormal?
- 3. IIIB: Zipf’s law. The authors use the symbolic transcription for Zipf’s law. Could the audio recording be used instead? One can envisage the comparison of two signals to see if they correspond to the same word, if they are close enough in their characteristics (correcting for different durations and physical frequency). May be the information provided in the corpus does not allow to do this, but in any case, it should be mentioned in the manuscript.
- 4. In the last years there has been considerable controversy about the fitting of power-law distributions. The controversy was initiated by Clauset et al. (SIAM Rev. 51, 661, 2009). But other authors have criticized Clauset et al. (Mainly Deluca and Corral, Acta Geophys. 61, 1351, 2013; Voitalov et al., arXiv:1811.02071; this is in the complex-systems literature, in the field of extreme-value statistics there is additional literature). The authors mention that they use maximum likelihood estimation, without any further cite to the way of selecting the cut-off of the power law (this is the key point). They do not mention either goodness-of-fit test. From my point of view, as this is a sort of generic, overview paper, it is not necessary that the authors take part in the fitting-of-power-laws controversy (more taking into account that the controversy is not solved, in my opinion). But it should be mentioned in the manuscript that the controversy exists, and that the authors leave for a future study the use of a more accurate fitting method. In this regard, the value $r=49$ separating the two found power-law regimes seems a bit ad-hoc. The authors should verify that other values of r , let us say, between 40 and 60 lead to very similar power-law exponents.
- 5. IIIC: Herdan’s law. As the corpus is a mixture of many different speakers it is not clear to me which are the effects of this, in particular the effect of randomness. Does the permutation of the

order of the different speakers change Herdan's law? What about a total randomization of the words? What about sorting the speakers in such a way that those slower go first and those fast at the end? And also in the opposite way. Discuss the results in the paper or mention as a future research, please.

- 6. IIID: Brevity law. It is not clear why the fits fit the median of the points and not the mean. As in the rest of the paper the mean is preferred in front of the median, this should be recognized. I am not asking for an explanation, just to mention that the median leads to much better results.

Explanation can be provided in future works.

- 7. IIIE: MA law. No justification is provided for the model given by Eq. 8. It would be good that the readers be informed which is the intuition behind that model, otherwise, it looks like an ad-hoc assumption. Which values of k_1 and k_2 lead to correlation? Positive or negative?

- 8. Further, are the authors sure the 2nd regime of the MA law is not an artifact, or spurious? In Fig. 11 the increase for high number of words only affect the last 2 points, and no error bars are provided to decide if the increase is significant. If these 2 points were removed probably the parameter c would be impossible to constrain. In Fig. 12 there is no increase at all.

- 9. General: Reproducibility and good documentation is nowadays important. The authors use several well-known tools (Kolmogorov-Smirnov, Levenberg-Marquardt, MLE, and other statistical tests, for instance) but do not cite which routines they are using. The routine, the programming language and the version should be stated in the paper (imagine that some day somebody finds a bug in one of those routines...). Also specify what is R^2 .

Other relevant points:

- 10. Several times the authors mention a debate between a "symbolic hypothesis" and a "physical hypothesis". However, no references are provided (at the end, they quote Refs. 31 and 81, but I haven't found these terms there). As this is an important point in the conclusions, the two hypotheses should be briefly explained. Are they the only possible hypotheses involved?

- 11. It is not clear to me how the physical units are defined. Is that something that is done in the corpus, so that the users do not need to care? An the authors change those definitions, or is that out of their control? Please mention in the text.

- 12. The authors mention that they find heavy-tailed distributions. Please note that one needs to be precise regarding this term, it is not the same heavy-tailed, long-tailed, subexponential, or regularly varying, see Voitalov et al., arXiv:1811.02071.

- 13. Figures 2, 3, 4 and 5 should be displayed as well with logarithmic vertical axes, to see clearly how the tail behaves. If there are deviations between the fits and the data, that's not important at this point, one would conclude then that the fits fit the main part of the distribution but not the tails. It is important to know the limitations.

- 14. The authors perform model selection by the minimization of the Kolmogorov-Smirnov distance. Could they provide a reference to justify such criterion? For instance, Clauset et al. use that criterion to fit, but not to choose between different models.

- 15. I disagree that the sum of 6 random variables is "very far away for the large n limit where CLT holds". The authors can verify that for 6 random variables (with a non-pathological initial distribution) the sum is close enough to a normal distribution.

- 16. In IIIA, in the model with non-independence, in fact, the authors do not perform any simulations. Please mention in the manuscript why. I guess it is because they do not have a model that describes the dependence, but this should be clear to the reader as well.

- 17. In Eq. (2) the authors use mutual information to quantify correlations. But why do they avoid the use of Pearson or Spearman correlation coefficients? In other parts of the manuscript they use Spearman, so, it seems a bit ad-hoc which tool to use. Please clarify in text.

- 18. Fig. 8. Does the p-value correspond to a test in which the Spearman correlation is zero? Which tests and software are used?

- 19. In Fig. 9: why the brevity law for phonemes is measured in terms of duration but not in terms in number of characters (as done in Fig. 8 for words)?

- 20. Mathematical formulation of the brevity law (III.D). Please explain what an alphabet in this

context is.

- 21. Figure 10: why is $r < 50$ excluded? If the results are not good, please mention, there is nothing wrong with that (not everything can be perfect).
- 22. III E: after Eq. 7. Please mention that the equation makes sense for negative b and c (it is unfortunate that they are negative, but I assume this is not the authors' fault).
- 23. Page 10: "the model... shows that there are actually two free parameters". Be careful, I think this enters into the category of fallacies: the authors find two parameters because their model has two parameters. A three-parameter model would lead to three parameters in the final formula. What the authors find is that two parameters suffice to fit the data, that's all.
- 24. Figure 12: why phoneme size in characters is not studied? Please clarify.
- 25. The last paragraph of Subsection III E explains an important result in a very summarized way, which makes it difficult to understand. I think the result deserves its own subsection, with more details.
- 26. In the same paragraph, the argument about efficiency does not seem right, as the limit is derived from the Buckeye corpus, so, it does not make sense that the data do not reach the limit of efficiency because the speakers speak in a relaxed way, when the limit of efficiency is derived for relaxed speakers.
- 27. In the discussion: "Since oral communication predates written communication, it sensible to wonder whether the emergence of linguistic laws in the latter context are indeed just a consequence of their emergence in the former." This statement cannot be so general, it would depend on the law. Laws for which time is important (such as the brevity law) are different to timeless laws (such as Zipf's law).
- 28. In the 3rd paragraph of the discussion, regarding the 2 scaling regimes, in this context it is important to take into account the results of Williams et al., Phys. Rev. E 91 052811 (2015). Multiauthor or multispeaker corpora have the problems mentioned there.
- 29. In the discussion, page 12: "we introduce for the first time a mathematical formulation of this law..., which follows from optimal compression principles from information theory..." It is not clear to me the value of such derivation, can it be considered a complete explanation? Or just a justification? The introduction of the prefactor λ_D measuring deviation from optimality seems somehow ad-hoc. Can one envisage a measurement to test if the information-theory optimization is indeed playing a role? I mean, even if the macroscopic outcome is correct, the underlying microscopic behavior can be different.
- 30. Can BGs be measured (automatically) in written texts? An explanation about BGs would be helpful in the manuscript.

Minor points:

- 31. BGs are sometimes referred to as Breath Groups and some others as Breathe Groups. Please unify terminology.
- 32. Zipf's law is sometimes referred to as Zipf law. Please correct.
- 33. Table 2. The value 0.449 for the gamma case seems too big (the maximum possible value is 1). That would be a terrible fit. Could the authors verify this is not a mistake?
- 34. Figure 8, caption. Correct: "...in all the cases cases)"
- 35. Page 9: "\ell is some property of the time duration distribution". I guess \ell has to be a measure of centrality of such distribution.
- 36. Please provide a reference for the harmonic numbers and the resulting formula.
- 37. Why are error bars missing in figures, from Fig. 8 to 12? It does not seem that they are going to be small. Please clarify in the manuscript why they are ignored.
- 38. Page 11: Please correct: "with the same the number of phonemes"

I apologize that I haven't read the supplementary information. In fact, when I downloaded the paper I didn't realized there was such information. I believe another referee has to review that before publication. I apologize also for my many comments, but I feel all of them are more or less relevant. And most of them are straightforward to deal with.

In summary, my recommendation is publication after the authors have addressed my comments in one way or another. I congratulate the authors for their hard work.

Review form: Reviewer 2

Is the manuscript scientifically sound in its present form?

Yes

Are the interpretations and conclusions justified by the results?

Yes

Is the language acceptable?

Yes

Do you have any ethical concerns with this paper?

No

Have you any concerns about statistical analyses in this paper?

No

Recommendation?

Accept as is

Comments to the Author(s)

The authors present a detailed, rigorous and extensive study of the four main linguistic statistical laws in oral communication, a matter that has been greatly overlooked in the literature so far. The manuscript is well written and easy to read. I did not find any methodological issues from the statistical analysis point of view.

Combining careful data analysis of the Buckeye Corpus with stochastic generative models, the authors are able to relate statistical regularities of oral and written communication. This, in turn, opens an interesting debate on the true origin of complexity in language: does it stem from its symbolic representation, or has it a prior, non-symbolic origin? While the research presented in this manuscript is in my opinion insufficient to give a definite answer, the authors give their view on this subject only in the closing remarks, and make it clear that further research is needed. Overall, I believe this manuscript can be published in its present form.

My only criticism would be that the authors do not provide access to their data and/or code used for the analysis. It would be great if the authors would facilitate reproducibility of their results by providing access to raw/processed data and code/scripts they used, to the extent that that is possible.

Decision letter (RSOS-191023.R0)

03-Jul-2019

Dear Mr González Torre

On behalf of the Editors, I am pleased to inform you that your Manuscript RSOS-191023 entitled "On the physical origin of linguistic laws and lognormality in speech" has been accepted for publication in Royal Society Open Science subject to minor revision in accordance with the referee suggestions. Please find the referees' comments at the end of this email.

The reviewers and handling editors have recommended publication, but also suggest some minor revisions to your manuscript. Therefore, I invite you to respond to the comments and revise your manuscript.

- Ethics statement

- Data accessibility

<http://datadryad.org/submit?journalID=RSOS&manu=RSOS-191023>

- Competing interests

- Authors' contributions

AB carried out the molecular lab work, participated in data analysis, carried out sequence alignments, participated in the design of the study and drafted the manuscript; CD carried out

the statistical analyses; EF collected field data; GH conceived of the study, designed the study, coordinated the study and helped draft the manuscript. All authors gave final approval for publication.

- Acknowledgements

- Funding statement

Because the schedule for publication is very tight, it is a condition of publication that you submit the revised version of your manuscript before 12-Jul-2019. Please note that the revision deadline will expire at 00.00am on this date. If you do not think you will be able to meet this date please let me know immediately.

- 1) A text file of the manuscript (tex, txt, rtf, docx or doc), references, tables (including captions) and figure captions. Do not upload a PDF as your "Main Document";
- 2) A separate electronic file of each figure (EPS or print-quality PDF preferred (either format should be produced directly from original creation package), or original software format);
- 3) Included a 100 word media summary of your paper when requested at submission. Please ensure you have entered correct contact details (email, institution and telephone) in your user account;
- 4) Included the raw data to support the claims made in your paper. You can either include your data as electronic supplementary material or upload to a repository and include the relevant doi within your manuscript. Make sure it is clear in your data accessibility statement how the data can be accessed;

5) All supplementary materials accompanying an accepted article will be treated as in their final form. Note that the Royal Society will neither edit nor typeset supplementary material and it will be hosted as provided. Please ensure that the supplementary material includes the paper details where possible (authors, article title, journal name).

on behalf of Dr Matjaz Perc (Associate Editor) and Miles Padgett (Subject Editor)
openscience@royalsociety.org

Reviewer comments to Author:

Reviewer: 1

Comments to the Author(s)

I am very much in favor of the publication of this paper. It is not only a great but an ambitious paper. The authors study the fulfilment in recorded speech of the 4 main laws of quantitative linguistics. In fact, they include an additional law, which describes the probability distribution of durations of words, phonemes, and breath groups (I do not know why they only mention “four laws” then). So, the scope of the paper is very broad, which is welcome. However, this broadness can be also a weak point, as the reader may find that some additional explanations and/or investigations can be necessary, to reach an in-depth treatment of each law. I am not in favor of “salami publication”, but the point is that, the more things in the paper, the more likely the referee finds something he/she doesn’t like.

My point is that this paper has to be considered a sort of “overview” and that in future research

the authors can extend their results. Nevertheless, it would be necessary that the authors mention in the manuscript that extensions and more in-depth analysis can be done, leaving the issue as an open one. So, it can be good for the readers to know that the research can be extended, and that the authors show the way to do that. It is within this philosophy that some of my comments below should be understood.

Most relevant points:

- 1. IIIA: Time durations: It is not totally clear what the authors are measuring. Please clarify in manuscript. Are they providing the distribution of all the individual events in the corpus (e.g., $3e5$ words)? Or the distribution of the mean value of each unique event (e.g., dictionary words)? Or is the median value instead of the mean? In any case, the former has of course greater variability. In other words, how are the events weighted? Any of these distributions is of interest, and they are related.
- 2. Related to this, my understanding is that time durations mix different speakers. Which is the effect of the velocity of every speaker? In other words, does the skewness come from the fact that they look at a diverse population of speakers? A single speaker also leads to a lognormal?
- 3. IIIB: Zipf's law. The authors use the symbolic transcription for Zipf's law. Could the audio recording be used instead? One can envisage the comparison of two signals to see if they correspond to the same word, if they are close enough in their characteristics (correcting for different durations and physical frequency). May be the information provided in the corpus does not allow to do this, but in any case, it should be mentioned in the manuscript.
- 4. In the last years there has been considerable controversy about the fitting of power-law distributions. The controversy was initiated by Clauset et al. (SIAM Rev. 51, 661, 2009). But other authors have criticized Clauset et al. (Mainly Deluca and Corral, Acta Geophys. 61, 1351, 2013; Voitalov et al., arXiv:1811.02071; this is in the complex-systems literature, in the field of extreme-value statistics there is additional literature). The authors mention that they use maximum likelihood estimation, without any further cite to the way of selecting the cut-off of the power law (this is the key point). They do not mention either goodness-of-fit test. From my point of view, as this is a sort of generic, overview paper, it is not necessary that the authors take part in the fitting-of-power-laws controversy (more taking into account that the controversy is not solved, in my opinion). But it should be mentioned in the manuscript that the controversy exists, and that the authors leave for a future study the use of a more accurate fitting method. In this regard, the value $r=49$ separating the two found power-law regimes seems a bit ad-hoc. The authors should verify that other values of r , let us say, between 40 and 60 lead to very similar power-law exponents.
- 5. IIIC: Herdan's law. As the corpus is a mixture of many different speakers it is not clear to me which are the effects of this, in particular the effect of randomness. Does the permutation of the order of the different speakers change Herdan's law? What about a total randomization of the words? What about sorting the speakers in such a way that those slower go first and those fast at the end? And also in the opposite way. Discuss the results in the paper or mention as a future research, please.
- 6. IIID: Brevity law. It is not clear why the fits fit the median of the points and not the mean. As in the rest of the paper the mean is preferred in front of the median, this should be recognized. I am not asking for an explanation, just to mention that the median leads to much better results. Explanation can be provided in future works.
- 7. IIIE: MA law. No justification is provided for the model given by Eq. 8. It would be good that the readers be informed which is the intuition behind that model, otherwise, it looks like an ad-hoc assumption. Which values of k_1 and k_2 lead to correlation? Positive or negative?
- 8. Further, are the authors sure the 2nd regime of the MA law is not an artifact, or spurious? In Fig. 11 the increase for high number of words only affect the last 2 points, and no error bars are provided to decide if the increase is significant. If these 2 points were removed probably the parameter c would be impossible to constrain. In Fig. 12 there is no increase at all.

- 9. General: Reproducibility and good documentation is nowadays important. The authors use several well-known tools (Kolmogorov-Smirnov, Levenberg-Marquardt, MLE, and other statistical tests, for instance) but do not cite which routines they are using. The routine, the programming language and the version should be stated in the paper (imagine that some day somebody finds a bug in one of those routines...). Also specify what is R^2 .

Other relevant points:

- 10. Several times the authors mention a debate between a “symbolic hypothesis” and a “physical hypothesis”. However, no references are provided (at the end, they quote Refs. 31 and 81, but I haven’t found these terms there). As this is an important point in the conclusions, the two hypotheses should be briefly explained. Are they the only possible hypotheses involved?
- 11. It is not clear to me how the physical units are defined. Is that something that is done in the corpus, so that the users do not need to care? An the authors change those definitions, or is that out of their control? Please mention in the text.
- 12. The authors mention that they find heavy-tailed distributions. Please note that one needs to be precise regarding this term, it is not the same heavy-tailed, long-tailed, subexponential, or regularly varying, see Voitalov et al., arXiv:1811.02071.
- 13. Figures 2, 3, 4 and 5 should be displayed as well with logarithmic vertical axes, to see clearly how the tail behaves. If there are deviations between the fits and the data, that’s not important at this point, one would conclude then that the fits fit the main part of the distribution but not the tails. It is important to know the limitations.
- 14. The authors perform model selection by the minimization of the Kolmogorov-Smirnov distance. Could they provide a reference to justify such criterion? For instance, Clauset et al. use that criterion to fit, but not to choose between different models.
- 15. I disagree that the sum of 6 random variables is “very far away for the large n limit where CLT holds”. The authors can verify that for 6 random variables (with a non-pathological initial distribution) the sum is close enough to a normal distribution.
- 16. In IIIA, in the model with non-independence, in fact, the authors do not perform any simulations. Please mention in the manuscript why. I guess it is because they do not have a model that describes the dependence, but this should be clear to the reader as well.
- 17. In Eq. (2) the authors use mutual information to quantify correlations. But why do they avoid the use of Pearson or Spearman correlation coefficients? In other parts of the manuscript they use Spearman, so, it seems a bit ad-hoc which tool to use. Please clarify in text.
- 18. Fig. 8. Does the p -value correspond to a test in which the Spearman correlation is zero? Which tests and software are used?
- 19. In Fig. 9: why the brevity law for phonemes is measured in terms of duration but not in terms in number of characters (as done in Fig. 8 for words)?
- 20. Mathematical formulation of the brevity law (III.D). Please explain what an alphabet in this context is.
- 21. Figure 10: why is $r < 50$ excluded? If the results are not good, please mention, there is nothing wrong with that (not everything can be perfect).
- 22. III E: after Eq. 7. Please mention that the equation makes sense for negative b and c (it is unfortunate that they are negative, but I assume this is not the authors’ fault).
- 23. Page 10: “the model... shows that there are actually two free parameters”. Be careful, I think this enters into the category of fallacies: the authors find two parameters because their model has two parameters. A three-parameter model would lead to three parameters in the final formula. What the authors find is that two parameters suffice to fit the data, that’s all.
- 24. Figure 12: why phoneme size in characters is not studied? Please clarify.
- 25. The last paragraph of Subsection III E explains an important result in a very summarized way, which makes it difficult to understand. I think the result deserves its own subsection, with more details.
- 26. In the same paragraph, the argument about efficiency does not seem right, as the limit is derived from the Buckeye corpus, so, it does not make sense that the data do not reach the limit

of efficiency because the speakers speak in a relaxed way, when the limit of efficiency is derived for relaxed speakers.

- 27. In the discussion: "Since oral communication predates written communication, it sensible to wonder whether the emergence of linguistic laws in the latter context are indeed just a consequence of their emergence in the former." This statement cannot be so general, it would depend on the law. Laws for which time is important (such as the brevity law) are different to timeless laws (such as Zipf's law).

- 28. In the 3rd paragraph of the discussion, regarding the 2 scaling regimes, in this context it is important to take into account the results of Williams et al., Phys. Rev. E 91 052811 (2015).

Multiauthor or multispeaker corpora have the problems mentioned there.

- 29. In the discussion, page 12: "we introduce for the first time a mathematical formulation of this law..., which follows from optimal compression principles from information theory..." It is not clear to me the value of such derivation, can it be considered a complete explanation? Or just a justification? The introduction of the prefactor λ_D measuring deviation from optimality seems somehow ad-hoc. Can one envisage a measurement to test if the information-theory optimization is indeed playing a role? I mean, even if the macroscopic outcome is correct, the underlying microscopic behavior can be different.

- 30. Can BGs be measured (automatically) in written texts? An explanation about BGs would be helpful in the manuscript.

Minor points:

- 31. BGs are sometimes referred to as Breath Groups and some others as Breathe Groups. Please unify terminology.

- 32. Zipf's law is sometimes referred to as Zipf law. Please correct.

- 33. Table 2. The value 0.449 for the gamma case seems too big (the maximum possible value is 1). That would be a terrible fit. Could the authors verify this is not a mistake?

- 34. Figure 8, caption. Correct: "...in all the cases cases)"

- 35. Page 9: "\ell is some property of the time duration distribution". I guess \ell has to be a measure of centrality of such distribution.

- 36. Please provide a reference for the harmonic numbers and the resulting formula.

- 37. Why are error bars missing in figures, from Fig. 8 to 12? It does not seem that they are going to be small. Please clarify in the manuscript why they are ignored.

- 38. Page 11: Please correct: "with the same the number of phonemes"

I apologize that I haven't read the supplementary information. In fact, when I downloaded the paper I didn't realized there was such information. I believe another referee has to review that before publication. I apologize also for my many comments, but I feel all of them are more or less relevant. And most of them are straightforward to deal with.

In summary, my recommendation is publication after the authors have addressed my comments in one way or another. I congratulate the authors for their hard work.

Reviewer: 2

Comments to the Author(s)

The authors present a detailed, rigorous and extensive study of the four main linguistic statistical laws in oral communication, a matter that has been greatly overlooked in the literature so far. The manuscript is well written and easy to read. I did not find any methodological issues from the statistical analysis point of view.

Combining careful data analysis of the Buckeye Corpus with stochastic generative models, the authors are able to relate statistical regularities of oral and written communication. This, in turn,

opens an interesting debate on the true origin of complexity in language: does it stem from its symbolic representation, or has it a prior, non-symbolic origin? While the research presented in this manuscript is in my opinion insufficient to give a definite answer, the authors give their view on this subject only in the closing remarks, and make it clear that further research is needed. Overall, I believe this manuscript can be published in its present form.

My only criticism would be that the authors do not provide access to their data and/or code used for the analysis. It would be great if the authors would facilitate reproducibility of their results by providing access to raw/processed data and code/scripts they used, to the extent that that is possible.

Author's Response to Decision Letter for (RSOS-191023.R0)

See Appendix A.

Decision letter (RSOS-191023.R1)

23-Jul-2019

Dear Mr González Torre,

I am pleased to inform you that your manuscript entitled "On the physical origin of linguistic laws and lognormality in speech" is now accepted for publication in Royal Society Open Science.

on behalf of Professor Matjaz Perc (Associate Editor) and Miles Padgett (Subject Editor)
openscience@royalsociety.org

Associate Editor Comments to Author (Professor Matjaz Perc):

Thank you for the comprehensive revision of your manuscript, which we are happy to accept for publication in Royal Society Open Science.

Appendix A

Dear editor,

we are very glad to know that both referees are very much in favor of the publication of this paper (the first referee after minor changes, the second as it stands), and to receive a similar positive evaluation by you. According to your suggestion, we hereby resubmit our paper where all the comments and suggestions from both referees have been thoroughly addressed in the revised version.

All changes have been highlighted in red in the manuscript for your and the referees perusal, and a list of the main modifications include:

- Reworded, added further details, explanations, mathematical derivations and references to address several points raised by referee 1.
- We have added several new analysis and figures (most of them in the SI) to address and solve some of the queries from referee 1. All the conclusions of our work hold.
- We have added all the processed data and scripts to two repositories for free access, and we have added such information in the manuscript, according to referee 1 and 2 suggestion.
- Added a more profound discussion on the dichotomy of the symbolic vs physical hypothesis.

In what follows we also add a detailed response to the comments of both referees. Comments by the referees are highlighted in blue, and our responses are provided in black.

We hope that this new version is now acceptable for publication in the journal.

Sincerely,
the authors.

Response to Reviewer 1

We would like to thank this referee for the **enormously** helpful comments and suggestions, which have been taken very thoroughly in our resubmission. We believe the paper has greatly improved. Below please find some explanations and responses to all his/her suggestions.

I am very much in favor of the publication of this paper. It is not only a great but an ambitious paper. The authors study the fulfilment in recorded speech of the 4 main laws of quantitative linguistics. In fact, they include an additional law, which describes the probability distribution of durations of words, phonemes, and breath groups (I do not know why they only mention “four laws” then).

AUTHORS: thank you for your assessment. We indeed agree with the reviewer and, certainly, we could speak of “five laws”, adding the relative one to the time duration distribution of linguistic elements (“lognormality law”), a contribution of the paper. Note that we also define yet another one! (the rank-size law), if we originally spoke about the four laws it was to follow the tradition of quantitative linguistics. In this new version we have clarified these aspects.

So, the scope of the paper is very broad, which is welcome. However, this broadness can be also a weak point, as the reader may find that some additional explanations and/or investigations can be necessary, to reach an in-depth treatment of each law. I am not in favor of “salami publication”, but the point is that, the more things in the paper, the more likely the referee finds something he/she doesn't like.

My point is that this paper has to be considered a sort of “overview” and that in future research the authors can extend their results. Nevertheless, it would be necessary that the authors mention in the manuscript that extensions and more in-depth analysis can be done, leaving the issue as an open one. So, it can be good for the readers to know that the research can be extended, and that the authors show the way to do that. It is within this philosophy that some of my comments below should be understood.

AUTHORS: Again we agree with the reviewer. It is true that the article covers many different angles (mathematical derivation/explanation of some laws, empirical ratification, proposal of new laws, etc). Despite the extent of our paper, we consider all angles to be entangled and related to each other, this is the reason why we considered better to compile a single paper instead of a list of them (a salami publication, as the referee says). Of course in each of the angles and topics we focus, we are not saying the final word, and several aspects are open problems: in this new submission we have tried to highlight some of these all over the paper and in the discussion section. As a matter of fact, we thank the referee for depicting such a long list (38) of comments, many of which indeed are relevant and interesting aspects which we comment in the discussion.

Most relevant points:

- 1. IIIA: Time durations: It is not totally clear what the authors are measuring. Please clarify in manuscript. Are they providing the distribution of all the individual events in the corpus (e.g., 3e5 words)? Or the distribution of the mean value of each unique event (e.g., dictionary words)? Or is the median value instead of the mean? In any case, the former has of course greater variability. In other words, how are the events weighted? Any of these distributions is of interest, and they are related.

AUTHORS: In this particular paper we are not displaying the time distribution of each different word or each different phoneme: this is a very fine-grained resolution which we leave for a future work (we are already working on this, but the paper would be way too long). We now comment on this aspect and clarify better. In section IIIA, what we do is collect the time of each token for all the types that we find, i.e. we are aggregating the time duration and displaying the time duration distribution of all the words and all the phonemes (and all BGs).

In other sections (e.g. Brevity Law) we consider the median value of the time of each specific word (this is explained in the text and the figures).

- 2. Related to this, my understanding is that time durations mix different speakers. Which is the effect of the velocity of every speaker? In other words, does the skewness come from the fact that they look at a diverse population of speakers? A single speaker also leads to a lognormal?

AUTHORS: Yes, we mix different speakers. Also, the models proposed do not require the data to be multi-speaker. We have extracted the time duration distributions computed on the data from 9 individual speakers, and the lognormal law still holds (this new information is included in the SI figure S1 and referenced in the main and the discussion).

- 3. IIIB: Zipf's law. The authors use the symbolic transcription for Zipf's law. Could the audio recording be used instead? One can envisage the comparison of two signals to see if they correspond to the same word, if they are close enough in their characteristics (correcting for different durations and physical frequency). May be the information provided in the corpus does not allow to do this, but in any case, it should be mentioned in the manuscript.

AUTHORS: This is a bit of a tour de force. To proceed as the referee suggests, first of all we would need a way of segmenting the signal (before pairs of signals could be compared), and this problem is in general very difficult if there is no "dictionary". So we would need to have a dictionary of signals for each word. This opens another problem: for instance, the word 'hello' can be spoken in many different ways, with different intonations, prosody, etc, so we would need to have, only for this word, a large set of 'hello signals' in order to have a reasonably accurate segmentator.

Furthermore, since each word can be spoken in different ways (e.g. with different time durations and energy release), it is not obvious how to explore Zipf's law in physical units.

As a matter of fact, some of us attempted something similar in [Sci. Rep. 7, 43862 (2017)], although in that case segmentation was based on energy fluctuation with respect to a given threshold. This was an agnostic way of segmenting the signal (and Zipf's law emerged), but types and tokens were not *necessarily* related to true words (we called them voice events).

Finally, an alternative solution would be to identify a spoken word with its time duration, assuming that two words are the same if the duration of the signal is the same (this is a very strong assumption which is probably wrong). Under that assumption, the time duration distribution of words (studied in section IIIA) would be the equivalent of a Zipf-like analysis (and we find lognormal distribution instead of a power-law relation in that case).

We have included some of this in the discussion section.

- 4. In the last years there has been considerable controversy about the fitting of power-law distributions. The controversy was initiated by Clauset et al. (SIAM Rev. 51, 661, 2009). But other authors have criticized Clauset et al. (Mainly Deluca and Corral, Acta Geophys. 61, 1351, 2013; Voitalov et al., arXiv:1811.02071; this is in the complex-systems literature, in the field of extreme-value statistics there is additional literature). The authors mention that they use maximum likelihood estimation, without any further cite to the way of selecting the cut-off of the power law (this is the key point). They do

not mention either goodness-of-fit test. From my point of view, as this is a sort of generic, overview paper, it is not necessary that the authors take part in the fitting-of-power-laws controversy (more taking into account that the controversy is not solved, in my opinion). But it should be mentioned in the manuscript that the controversy exists, and that the authors leave for a future study the use of a more accurate fitting method. In this regard, the value $r=49$ separating the two found power-law regimes seems a bit ad-hoc. The authors should verify that other values of r , let us say, between 40 and 60 lead to very similar power-law exponents.

AUTHORS: We have acknowledged the existing controversy regarding power law fits in the new version of the manuscript, along with two new references (to Clauset et al and to Deluca and Corral).

All fits in the paper have been done using MLE and goodness of fits (e.g. for the fits of time duration distribution of phonemes, words and BGs to different candidate distributions such as lognormal, Gamma, Weibull, etc) have been assessed via Kolmogorov-Smirnov distance.

Note that the power law fit mainly applies to the case of Zipf's law. In the case of Zipf's law for words, we have not used any cut-off: the value of $r=49$ is now hand-picked, it is the one that maximize the likelihood of the two regime power-law to data (so basically, we have used MLE to find $r=49$). This is now explained in the text.

As mentioned by the reviewer, small variations in the selection of r leads to small variations in the regime exponents. Selecting a cut-off point for the first regime would also lead to small variations of the exponents and r . In this case we have not used any goodness-of-fit and we have not tried to fit any other function as it was not the main intention of this analysis.

- 5. IIIC: Herdan's law. As the corpus is a mixture of many different speakers it is not clear to me which are the effects of this, in particular the effect of randomness. Does the permutation of the order of the different speakers change Herdan's law? What about a total randomization of the words? What about sorting the speakers in such a way that those slower go first and those fast at the end? And also in the opposite way. Discuss the results in the paper or mention as a future research, please.

AUTHORS: Thanks for the interesting comment. We have now produced 10 permutations of the ordering and computed Herdan's law in each permutation, and have replaced the old figure (with just one permutation) with all the curves. Transient is affected mainly for the version of the law where units is elapsed time, but the stable sublinear scaling regime is not affected by permutations, nor the exponents. We have discussed this aspect as well in the text.

- 6. IIID: Brevity law. It is not clear why the fits fit the median of the points and not the mean. As in the rest of the paper the mean is preferred in front of the median, this should be recognized. I am not asking for an explanation, just to mention that the median leads to much better results. Explanation can be provided in future works.

AUTHORS: For section IIIA (the law of lognormality), IIIB (Zipf's law) and IIIC (Herdan's law) there is not such confusion since we don't need to choose between mean and

median. For Menzerath-Altman law we have decided to use the mean because it is the “traditional” way of doing it. However, there is not many previous research for Brevity law in terms of oral units, so we suggest to use median instead of mean simply because we know that mean values are less informative due to the fact that time distributions are lognormal (heavy tailed).

- 7. IIIE: MA law. No justification is provided for the model given by Eq. 8. It would be good that the readers be informed which is the intuition behind that model, otherwise, it looks like an ad-hoc assumption. Which values of k_1 and k_2 lead to correlation? Positive or negative?

AUTHORS: Thanks for pointing this out. In this version we have added additional intermediate steps to justify the meaning of $kappa_1$ and $kappa_2$ (both are positive, but $kappa_2 < 1$ and $kappa_1 > 1$). If we only use $kappa_2$ and remove $kappa_1$, then the model simply uses $kappa_2$ as a sort of correlation coefficient and by introducing it, we recover the restricted version of MAL. $kappa_1$ is an additional parameter that balances out $kappa_2$. All the new derivations and explanations are reported in the new version of the manuscript.

- 8. Further, are the authors sure the 2nd regime of the MA law is not an artifact, or spurious? In Fig. 11 the increase for high number of words only affect the last 2 points, and no error bars are provided to decide if the increase is significant. If these 2 points were removed probably the parameter c would be impossible to constrain. In Fig. 12 there is no increase at all.

AUTHORS: Note that the second regime is a mathematical prediction of the MAL itself (it simply has gone unnoticed!) One cannot be certain to which extent experimentally there are two regimes, just one, or neither, but the best fits to the binned data provide a set of parameters (a, b, c) which suggest the presence of the two regimes. Note that other works also found this experimentally –although without noticing the regime shift nor acknowledging this fact–, including the pioneering works of Menzerath and the revisions of Gabriel Altmann, as acknowledged in the paper. In any case, in the discussion section we open the question of whether the second regime is truly there and whether it is significant.

- 9. General: Reproducibility and good documentation is nowadays important. The authors use several well-known tools (Kolmogorov-Smirnov, Levenberg-Marquardt, MLE, and other statistical tests, for instance) but do not cite which routines they are using. The routine, the programming language and the version should be stated in the paper (imagine that some day somebody finds a bug in one of those routines...). Also specify what is R^2 .

AUTHORS: We now specify what R^2 is in table III caption (R^2 is the coefficient of determination between of the fitting to the blue bins).

We have also added a reproducibility statement, where we direct the reader to two different repositories where processed data and all the scripts have been allocated (note that the Buckeye Corpus is a freely-accessible corpus for non-commercial uses

available in its web page).

In this reproducibility statement we also depict software characteristics (we have used Python 3.7 for the analysis, Levenberg-Marquardt, Spearman test and most of MLE fits using Scipy 1.3.0; MLE fit for power law is self-coded and archived, and other libraries such as Numpy 1.16.2, Pandas 0.24.2 or Matplotlib 3.1.0 are also used).

Other relevant points:

- 10. Several times the authors mention a debate between a “symbolic hypothesis” and a “physical hypothesis”. However, no references are provided (at the end, they quote Refs. 31 and 81, but I haven’t found these terms there). As this is an important point in the conclusions, the two hypotheses should be briefly explained. Are they the only possible hypotheses involved?

AUTHORS: Thanks for this important comment. The specific wording ‘symbolic hypothesis vs physical hypothesis’ is ours, and we now acknowledge it in the text. The ‘symbolic hypothesis’ is a well-established hallmark that goes all the way back to Chomsky: we have now added references and a better framing of this paradigm in the introduction. The ‘physical hypothesis’ has a smaller and definitely more scattered background, in fact to the best of our knowledge this is the first paper where the general dichotomy is considered in the context of linguistic laws and the general conceptual paradigm is introduced. We have added more explanations and references in the discussion, and have tried to better explain this dichotomy.

- 11. It is not clear to me how the physical units are defined. Is that something that is done in the corpus, so that the users do not need to care? An the authors change those definitions, or is that out of their control? Please mention in the text.

AUTHORS: See table I. The only physical magnitude that we use in this paper is time (we could also consider energy, but that would make the paper way too long, we have added this remark in the discussion section). For section III, we consider the time duration of each token at each of the 3 linguistic levels (phoneme, word, BG), whereas for the sections on linguistic laws we typically consider averaged quantities (e.g. the mean or median time of each word, explained in the text).

- 12. The authors mention that they find heavy-tailed distributions. Please note that one needs to be precise regarding this term, it is not the same heavy-tailed, long-tailed, subexponential, or regularly varying, see Voitalov et al., arXiv:1811.02071.

AUTHORS: we have specified that we find these heavy-tailed are actually subexponential (lognormal) in the classification of Voitalov et al, and added a reference.

- 13. Figures 2, 3, 4 and 5 should be displayed as well with logarithmic vertical axes, to see clearly how the tail behaves. If there are deviations between the fits and the data, that’s not important at this point, one would conclude then that the fits fit the main part of the distribution but not the tails. It is important to know the limitations.

AUTHORS: We have added such plots in the SI. We find that for phonemes all the data are well fitted (including the tail), but for words and BGs the tail deviates a bit. Interestingly, our theoretical model fits better, even in the tails.

- 14. The authors perform model selection by the minimization of the Kolmogorov-Smirnov distance. Could they provide a reference to justify such criterion? For instance, Clauset et al. use that criterion to fit, but not to choose between different models.

AUTHORS: The referee is right, D_{KS} is not a criterion for model selection. Since all the models have the same number of parameters, we can directly use MLE to make model selection (because AIC or BIC for instance essentially reduce to comparing likelihood functions when all the models have the same parameters). In particular, we now compute the mean log-likelihood for each model (dividing the loglikelihood by the number of data), and the model to be selected is the one with higher mean log-likelihood. We have added this new data to table II (we keep the D_{ks} values as these still provide goodness of fit evidence).

Similarly as before, we find that for both phonemes and BG time durations, the lognormal model is the clear winner, whereas for words there is a tie between three models (however the lognormal is the only one which has associated a generative mechanism, i.e. Eq.1).

- 15. I disagree that the sum of 6 random variables is “very far away for the large n limit where CLT holds”. The authors can verify that for 6 random variables (with a non-pathological initial distribution) the sum is close enough to a normal distribution.

AUTHORS: This is not so clear-cut, actually there is substantial literature on the “moderate” sum of lognormal random variables don’t converging to CLT, see for instance Romeo et al, Broad distribution effects in sums of lognormal random variables, EPJB 32, 4 (2003). A reference-listing document summarizing the rule-of-thumb by which the sum of a few lognormals is well approximated by a lognormal is [Dufresne, Sums of lognormals, Actuarial Research Conference 2008], available at <http://ozdaniel.com/A/DufresneLognormalsARC2008.pdf>

For instance, it is easy to sum thousands of lognormal random variables and this sum being skewed enough for the normal limit to be rejected! See below for an example where we sum 1000 lognormals and the limiting distribution is **clearly not** Gaussian (left panel should look Gaussian!). The limit distribution of the sum is more similar to a lognormal (which looks Gaussian in linear-log scale, see right panel) although note that it is not quite lognormal as it is slightly skewed.

For a reasonable range of lognormal parameters similar to those found in the Buckeye corpus, we have checked that the sum of a small number of lognormals can be well approximated by a lognormal. Figure S1 (with 11 panels) in the supplementary material provides a glance of the rich taxonomy of the sum of a small number of lognormals.

- 16. In IIIA, in the model with non-independence, in fact, the authors do not perform any simulations. Please mention in the manuscript why. I guess it is because they do not have a model that describes the dependence, but this should be clear to the reader as well.

AUTHORS: Yes, we do indeed do these simulations at the time we introduce MA law (details and results are depicted in the SI figure S10), because it is in the MA section where we deal with dependencies between the words forming a BG etc. We clarify this in section IIIA. Note that in section III we don't need to explicitly add non-independence to find good reconstruction of time duration distributions, so this seems to be more of a second order effect for understanding lognormality (which nonetheless is key for the formation of linguistic laws such as MAL).

- 17. In Eq. (2) the authors use mutual information to quantify correlations. But why do they avoid the use of Pearson or Spearman correlation coefficients? In other parts of the manuscript they use Spearman, so, it seems a bit ad-hoc which tool to use. Please clarify in text.

AUTHORS: We cannot assume a priori any particular kind of dependence –Pearson and Spearman correlation coefficients assume linear and monotonic correlations–, just dependency, so we thought using mutual information which we considered a more general measure. We have added this clarification.

- 18. Fig. 8. Does the p-value correspond to a test in which the Spearman correlation is zero? Which tests and software are used?

AUTHORS: It is a for Spearman test: a two-sided p-value for a hypothesis test whose null hypothesis is that two sets of data are uncorrelated, A small p-value (typically ≤ 0.05) indicates strong evidence against the null hypothesis, so you reject the null hypothesis. Computed with Scipy.

- 19. In Fig. 9: why the brevity law for phonemes is measured in terms of duration but not in terms in number of characters (as done in Fig. 8 for words)?

AUTHORS: The Brevity law at the phoneme level cannot be unambiguously explored in terms of number of characters, simply because a given spoken word might very well be decomposed in different sets of phonemes when transcribed (see panel (b) in Figure 1 for an illustration). For instance when the phonetic transcription of an spoken instance of the word okay is 'nowkai', it is not clear how to assign characters to each of the phonemes in 'nowkai'. The problem is therefore messy and we have skipped it in this paper, but now we have acknowledged this limitation in the text.

- 20. Mathematical formulation of the brevity law (III.D). Please explain what an alphabet in this context is.

AUTHORS: The Alphabet here is the set of different linguistic elements, at the level of study analyzed (for instance if the alphabet is the set of letters, $D=26$, if it is the set of phonemes, $D=64$). We have now clarified this.

- 21. Figure 10: why is $r < 50$ excluded? If the results are not good, please mention, there is nothing wrong with that (not everything can be perfect).

AUTHORS: We just focused in the region where the second exponent of the Zipf law emerged. For $r < 50$ we needed to use the first exponent found in the double-power law, and a different 'straight line' shows up. In the new version we have changed this figure to introduce the additional range, and changed a bit the text to discuss this.

- 22. III E: after Eq. 7. Please mention that the equation makes sense for negative b and c (it is unfortunate that they are negative, but I assume this is not the authors' fault).

AUTHORS: We are using a standard formulation of the law. Note that b and c do not **necessarily** need to be < 0 . See below an example of MAL $y(n) = a n^b \exp(-cn)$, for different combinations of parameters and all of them showing the characteristic decay:

- 23. Page 10: "the model... shows that there are actually two free parameters". Be careful, I think this enters into the category of fallacies: the authors find two parameters

because their model has two parameters. A three-parameter model would lead to three parameters in the final formula. What the authors find is that two parameters suffice to fit the data, that's all.

AUTHORS: We agree and have rewritten that sentence.

- 24. Figure 12: why phoneme size in characters is not studied? Please clarify.

AUTHORS: A similar issue has been already addressed in the response to comment (19), here the reason why we haven't explored MAL with respect to phoneme size in characters is the same.

- 25. The last paragraph of Subsection III E explains an important result in a very summarized way, which makes it difficult to understand. I think the result deserves its own subsection, with more details.

AUTHORS: We have set this as a paragraph with a bold name, similar to what we do in other subsections, and have expanded a bit the explanation.

- 26. In the same paragraph, the argument about efficiency does not seem right, as the limit is derived from the Buckeye corpus, so, it does not make sense that the data do not reach the limit of efficiency because the speakers speak in a relaxed way, when the limit of efficiency is derived for relaxed speakers.

AUTHORS: We have probably not explained well the argument. MAL dictates that speakers will speak faster when the BG is larger until reaching a critical point which in the database is around $n^*=34$ words per BG, and for longer BGs this tendency is inverted. In our database, of course we find BGs of all sizes, and according to MAL that means we will find speakers whose speech velocity will vary, sometimes actually reaching its maximum (what we called the optimal efficiency limit, i.e. the speed at which one can maximize the number of spoken words/second). However, on **average** the BGs size notable smaller than the BG size for which (according MAL) the speech velocity is maximal, that means that on **average** the speech velocity of speakers is below this optimal limit. We argue that this can be due to different factors, including the fact that **on average (or typically)** the speaker chats in a relaxed environment, so not in need to maximize the number of spoken words/second. We have tried to clarify this point in the resubmission.

- 27. In the discussion: "Since oral communication predates written communication, it sensible to wonder whether the emergence of linguistic laws in the latter context are indeed just a consequence of their emergence in the former." This statement cannot be so general, it would depend on the law. Laws for which time is important (such as the brevity law) are different to timeless laws (such as Zipf's law).

AUTHORS: We would argue that Zipf's law is related to Brevity law (actually Zipf himself originally proposed these laws hand to hand), and is also related to Herdan's law (which again is time dependent in our context).

Nevertheless, the logic of our argument is as follows: oral communication emerged

before written communication. We have traditionally explored linguistic laws in written communication, but do they indeed arise there (e.g. due to symbolization) or do they already emerge in oral communication? In this argumentation line, it is equally sensible to explore time-dependent and time-independent laws.

- 28. In the 3rd paragraph of the discussion, regarding the 2 scaling regimes, in this context it is important to take into account the results of Williams et al., Phys. Rev. E 91 052811 (2015). Multiauthor or multispeaker corpora have the problems mentioned there.

AUTHORS: Thanks for this very interesting reference, which we now cite. We have computed Zipf's law for 9 individual speakers and, in apparent compliance with Williams et al, found that a double power law might not be needed there, so it is possible that the double power law scaling is a byproduct of multi-speaker nature of the corpus. We added this new evidence in the SI and discuss it in the main.

- 29. In the discussion, page 12: "we introduce for the first time a mathematical formulation of this law..., which follows from optimal compression principles from information theory..." It is not clear to me the value of such derivation, can it be considered a complete explanation? Or just a justification? The introduction of the prefactor λ_D measuring deviation from optimality seems somehow ad-hoc. Can one envisage a measurement to test if the information-theory optimization is indeed playing a role? I mean, even if the macroscopic outcome is correct, the underlying microscopic behavior can be different.

AUTHORS: We agree that the mathematical formulation is not a complete explanation as the arguments we use to build up the "microscopic" theory, while intuitively correct, have not been empirically checked. We have tried to nuance our claim on this aspect.

- 30. Can BGs be measured (automatically) in written texts? An explanation about BGs would be helpful in the manuscript.

AUTHORS: Not really a priori, BG is an oral magnitude. In written texts perhaps a correspondence could be established between text and pauses by performing experiments where subjects read aloud. Naïve indications of BGs could be driven by punctuation, however in spontaneous speech this is not that obvious. We briefly discuss this issue at the time we define breath groups.

Minor points:

- 31. BGs are sometimes referred to as Breath Groups and some others as Breathe Groups. Please unify terminology.

AUTHORS: corrected, with thanks (it's breath groups, not breathe groups, thanks for pointing this out).

- 32. Zipf's law is sometimes referred to as Zipf law. Please correct.

AUTHORS: corrected, with thanks

- 33. Table 2. The value 0.449 for the gamma case seems too big (the maximum possible value is 1). That would be a terrible fit. Could the authors verify this is not a mistake?

AUTHORS: Thanks for spotting this, there actually was a convergence issue for this case, we have checked and corrected this in the new version. Note that, additionally, table 2 has now been modified to include model selection parameters (mean loglikelihood).

- 34. Figure 8, caption. Correct: "...in all the cases cases)"

AUTHORS: corrected, with thanks.

- 35. Page 9: "\ell is some property of the time duration distribution". I guess \ell has to be a measure of centrality of such distribution.

AUTHORS: corrected, with thanks.

- 36. Please provide a reference for the harmonic numbers and the resulting formula.

AUTHORS: done, with thanks.

- 37. Why are error bars missing in figures, from Fig. 8 to 12? It does not seem that they are going to be small. Please clarify in the manuscript why they are ignored.

AUTHORS: Dots are the result of data binning, and as it is customary for e.g. when performing a logarithmic binning of an empirical power law distribution, error bars are not included. This is also common practice for e.g. analysis of Menzerath-Altmann law, we expect error bars to be significantly large (and one can see that from the raw data, plotted in background in the figure), but the law describes the average behavior (not the typical).

- 38. Page 11: Please correct: "with the same the number of phonemes"

AUTHORS: corrected, with thanks.

I apologize that I haven't read the supplementary information. In fact, when I downloaded the paper I didn't realized there was such information. I believe another referee has to review that before publication. I apologize also for my many comments, but I feel all of them are more or less relevant. And most of them are straightforward to deal with.

AUTHORS: We thank the referee for the extensive list of comments, that have indeed helped us to greatly improve the quality of the paper. Some elements regarding those issues are partially addressed in the SI, although the main paper stands on its own and the SI offers additional analysis and explanations supporting some of our claims.

In summary, my recommendation is publication after the authors have addressed my comments in one way or another. I congratulate the authors for their hard work.

AUTHORS: We thank the referee for these final words and hope that the changes made in this resubmission, along with the explanations conveyed in this letter, are sufficient to propose publication.

Response to Reviewer 2

The authors present a detailed, rigorous and extensive study of the four main linguistic statistical laws in oral communication, a matter that has been greatly overlooked in the literature so far. The manuscript is well written and easy to read. I did not find any methodological issues from the statistical analysis point of view.

Combining careful data analysis of the Buckeye Corpus with stochastic generative models, the authors are able to relate statistical regularities of oral and written communication. This, in turn, opens an interesting debate on the true origin of complexity in language: does it stem from its symbolic representation, or has it a prior, non-symbolic origin? While the research presented in this manuscript is in my opinion insufficient to give a definite answer, the authors give their view on this subject only in the closing remarks, and make it clear that further research is needed. Overall, I believe this manuscript can be published in its present form.

Authors: We are very glad to read this assessment.

My only criticism would be that the authors do not provide access to their data and/or code used for the analysis. It would be great if the authors would facilitate reproducibility of their results by providing access to raw/processed data and code/scripts they used, to the extent that that is possible.

AUTHORS: Buckeye corpus is a freely accessible corpus for non-commercial uses. Post-processed data from Buckeye corpus and all scripts have now been put in respective repositories for reproducibility, and cited in the manuscript.